# PROMPT GRADIENT PROJECTION FOR CONTINUAL LEARNING

**Jingyang Qiao**[1]*, **Zhizhong Zhang**[1]*, **Xin Tan**[1], **Chengwei Chen**[2], **Yanyun Qu**[3], **Yong Peng**[4], **Yuan Xie**[1]($\boxtimes$)

[1]East China Normal University, [2]The Navy Military Medical University,
[3]Xiamen University, [4]Central South University
`52275901010@stu.ecnu.edu.cn`, `{zzzhang,xtan}@cs.ecnu.edu.cn`
`timchen91@aliyun.com`, `yyqu@xmu.edu.cn`, `yong_peng@csu.edu.cn`
`yxie@cs.ecnu.edu.cn`

## ABSTRACT

Prompt-tuning has demonstrated impressive performance in continual learning by querying relevant prompts for each input instance, which can avoid the introduction of task identifier. Its forgetting is therefore reduced as this instance-wise query mechanism enables us to select and update only relevant prompts. In this paper, we further integrate prompt-tuning with gradient projection approach. Our observation is: prompt-tuning releases the necessity of task identifier for gradient projection method; and gradient projection provides theoretical guarantees against forgetting for prompt-tuning. This inspires a new **p**rompt **g**radient **p**rojection approach (PGP) for continual learning. In PGP, we deduce that reaching the orthogonal condition for prompt gradient can effectively prevent forgetting via the self-attention mechanism in vision-transformer. The condition equations are then realized by conducting Singular Value Decomposition (SVD) on an element-wise sum space between input space and prompt space. We validate our method on diverse datasets and experiments demonstrate the efficiency of reducing forgetting both in class incremental, online class incremental, and task incremental settings. The code is available at https://github.com/JingyangQiao/prompt-gradient-projection.

## 1 INTRODUCTION

Learning continually while not forgetting is a long-standing pursuit of machine learning systems (Kumaran et al., 2016; McClelland et al., 1995; Arani et al., 2022). Incremental learning, or continual learning is such a fabulous way to train a model with continuously expanded datasets by adding novel classes or domains (Ring, 1997; Hadsell et al., 2020; De Lange et al., 2021). In general, continual learning includes two distinct settings, *i.e.,* class- and task-incremental learning (Van de Ven & Tolias, 2019), abbreviated as CIL and TIL respectively. The main difference is whether the task identifier, *i.e.,* the samples belong to which training tasks, is given for inference.

Recently, the appearance of the prompt-tuning paradigm provides a new sight for class-incremental learning (Wang et al., 2022a; Li et al., 2023). In this framework, a tiny set of trainable tokens, *i.e.,* prompts, are combined with image features, and forwarded into a fixed Transformer architecture. As the instance-wise query mechanism can select relevant prompts according to the input sample, only these relevant parts are updated during training (Wang et al., 2022c). Since the idea of prompt-tuning is borrowed from the area of natural language processing (NLP) (Lester et al., 2021; Li & Liang, 2021), its deep mechanism against forgetting has not been revealed yet (Zhou et al., 2023).

Fortunately, it is observed that learning would not forget if the gradient is updated in the orthogonal direction to the subspace spanned by the old inputs, *i.e.,* gradient projection approaches (GP) (Saha et al., 2021). However, one obvious limitation is that GP is only applicable for task incremental learning, because the gradient constraints would greatly restrict the learning of new tasks compared with normal training (Zhao et al., 2023). Thus, it needs task identifier to instruct updating.

---

*Equal contribution

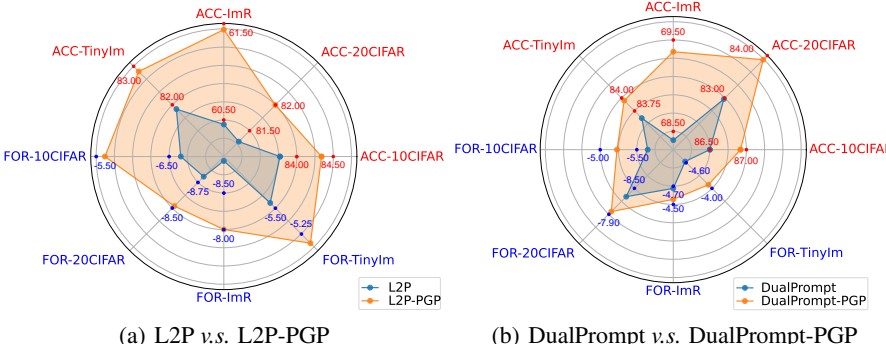

(a) L2P *v.s.* L2P-PGP        (b) DualPrompt *v.s.* DualPrompt-PGP

Figure 1: Radar chart of comparisons in terms of average accuracy and forgetting between baselines and our methods. L2P (Wang et al., 2022c) and DualPrompt (Wang et al., 2022b) are two state-of-the-art prompt-tuning approaches for continual learning. ACC refers to the average accuracy metric (higher is better). FOR refers to the forgetting metric (lower is better). Different scale standards are adopted for two metrics on benchmark datasets.

Based on this observation, we propose to combine prompt-tuning and gradient projection for further anti-forgetting. This combination enjoys: i) prompt-tuning with the instance-wise query mechanism releases the limitation of task identifier for gradient projection; ii) gradient projection provides the theoretical guarantees against forgetting for prompt-tuning.

In this paper, we propose a novel prompt gradient projection (PGP) for continual learning. We recall the pipeline of prompt-based continual learning (prompt-tuning) and deduce the orthogonal condition of anti-forgetting for prompt gradient via the self-attention mechanism in vision-transformer. We solve the condition equations by conducting Singular Value Decomposition (SVD) on an element-wise sum space between input space and prompt space. That allows us to obtain the gradient projection matrix in an efficient way.

We validate our approach in four benchmark datasets: CIFAR-100, ImageNet-R, TinyImageNet, and CUB200, with three baselines of L2P (Wang et al., 2022c), DualPrompt (Wang et al., 2022b), and CLIP (Radford et al., 2021), where an extraordinary anti-forgetting property is observed shown in Figure 1. We are the first to explicitly provide anti-forgetting mechanism for prompt-based continual learning, and hope our study will further inspire follow-up works. Our contributions are:

(1) Prompt gradient projection is the first work to study the anti-forgetting mechanism of prompt-tuning. Our approach obtains the orthogonal condition of anti-forgetting for prompt gradient and hence the retention of old knowledge has a rigorous theoretical guarantee.

(2) We provide a new viewpoint about stability and plasticity by investigating the selection of prompt gradient projection matrix. It appears that the essence of gradient projection is actually a trade-off, where the optimal solution is updating prompt in the orthogonal space of previous tasks.

(3) We apply our approach in both prompt-tuning and prefix-tuning paradigms and show the effectiveness of our approach. Our approach achieves state-of-the-art results in terms of forgetting metric and average accuracy metric, under the settings of class incremental learning, online class incremental learning, and task incremental learning.

## 2    RELATED WORKS AND PRELIMINARIES

Continual learning is defined as training deep neural networks (DNN) on time-variant data, *i.e.,* a sequence of tasks, marked as $\mathcal{D} = \{\mathcal{D}_1, ..., \mathcal{D}_T\}$, where $t$-th task $\mathcal{D}_t = \{(X_i^t, y_i^t)_{i=1}^{n_t}\}$ contains tuples of input sample $X_i^t \in \mathcal{X}_t$ and corresponding label $y_i^t \in \mathcal{Y}_t$. When a task $\mathcal{X}_t$ arrives, a model $f_\theta$ would be trained for the current task, while the data from previous tasks is unreachable. In this work, we mainly focus on class incremental learning, without knowing the task identifier during inference.

## 2.1 PROMPT-BASED CLASS INCREMENTAL LEARNING

A simple yet effective prompt-based (prompt-tuning) CIL model: Learning to Prompt (L2P) (Wang et al., 2022c) is first proposed. In it, prompt $p$, a tiny set of trainable tokens, combined with image features, are sent into vision-transformer, instructing the model to resist forgetting. To pick appropriate prompts for task-specific training, L2P deployed a prompt pool $P$ including plenty of prompt-key pairs, $\{p_j, k_j\}_{j=1}^{M}$, where $k_j$ represents the $j$-th key and $M$ is the total number of prompt-key pairs.

Based on L2P, DualPrompt (Wang et al., 2022b) divided the prompts into two parts: expert prompt and general prompt for distinct features learning. DualPrompt also replaced prompt-tuning with prefix-tuning, which has been successfully proven in the area of NLP. DyTox (Douillard et al., 2022) designed a novel task attention block, which utilized the task tokens to infer task identifier. Coda-Prompt (Smith et al., 2023) replaced the prompt pool with a decomposed prompt that consists of a weighted sum of learnable prompt components, allowing itself optimized in an end-to-end fashion with high plasticity. LGCL (Khan et al., 2023) introduced the text information into the learning of prompt pool, improving performance without any additional learnable parameters.

Although prompt-based CIL shows state-of-the-art performance, forgetting still exists compared with other incremental approaches (Saha et al., 2021). Since the problem of forgetting is not explicitly modeled in this framework, its mechanism against forgetting has not been revealed yet.

## 2.2 BACKGROUND OF GRADIENT PROJECTION METHOD

Gradient limitation, *i.e.,* restricting the gradient direction, originated from mathematical theory, provides an important explanation of the stability-plasticity dilemma (Kirkpatrick et al., 2017; Serra et al., 2018; Chaudhry et al., 2018; Farajtabar et al., 2020; Zeng et al., 2019; Wang et al., 2021).

Recent studies found that learning would not forget if the gradient is updated in the orthogonal direction of the subspace spanned by the old features. Gradient projection method (GPM) (Saha et al., 2021) updated the weights in the direction orthogonal to the subspace spanned by all previously learned inputs. This ensured that new learning processes did not interfere with previously learned tasks. Trust Region Gradient Projection (TRGP) (Lin et al., 2022b) selected old tasks in the trust region to learn new tasks by a layer-wise scaling matrix, together with orthogonal gradient projection. Simple Linear Connector (Connector) (Lin et al., 2022a) merged two models by using a weighted sum function where one model is updated normally and another is updated with gradient projection.

To further illustrate the anti-forgetting reason of gradient projection, we denote the inputs of task $t$ for layer $l$ as $S_t^l$, the learned model for task $t$ as $\{W_t^l\}_{l=1}^{L}$, and $L$ is the total number of layers. In the subsequent sections, we omit layer $L$ for simplicity. Let $\Delta W_t$ denote the model change after learning task $t+1$. If the update direction is orthogonal to the old features, it follows that $\Delta W_t x_{t,i} = 0$, and $x_{t,i} \in S_t$, where the index "$t, i$" means the $i$-th input image of task $t$ (Saha et al., 2021; Lin et al., 2022b). Therefore, as the model $W_{t+1}$ is updated as $W_{t+1} = W_t + \Delta W_t$, validating the performance of model $W_{t+1}$ on task $t$, we have:

$$W_{t+1}x_{t,i} = (W_t + \Delta W_t)x_{t,i} = W_t x_{t,i} + \Delta W_t x_{t,i} = W_t x_{t,i}, \tag{1}$$

which indicates that no interference is introduced to old tasks after learning a new concept, thereby addressing the forgetting issue.

However, one limitation of gradient projection methods that fail in the class-incremental inference is that the projected gradient needs task identifier to find relevant update parameters. In this paper, we will illustrate: prompt-tuning can break the constraints of needing **task identifier** in gradient projection and therefore the combination of prompt and gradient projection shows advanced properties in class incremental learning.

## 3 METHOD

The flowchart of our method is shown in Figure 2. In Figure 2(a), prompts are chosen according to the similarity between the key vector [1] and the query feature. The picked prompts are then

---

[1]In L2P, the key vector is initialized randomly in the form of one-dimension vector and trained to match the query feature of the corresponding task.

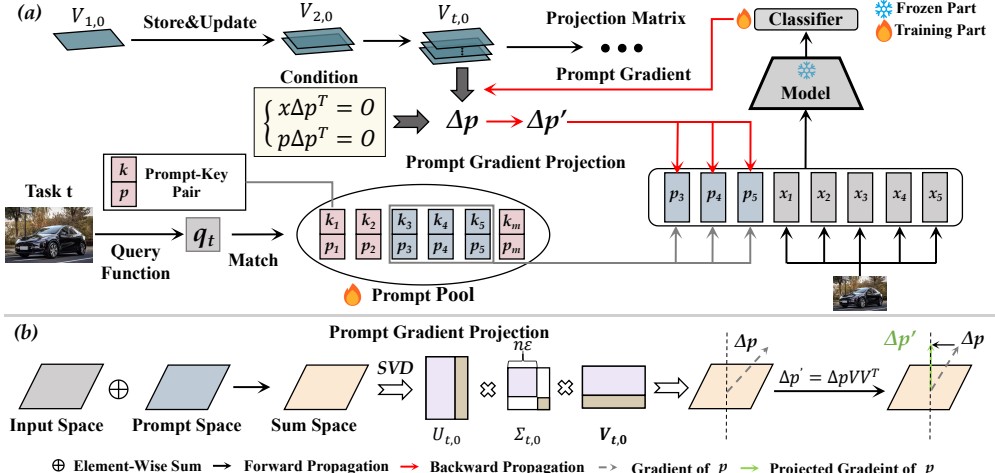

Figure 2: Flowchart of our work, (a) Process of forward/backward (black/red line). An instance-wise query mechanism is adopted during forward propagation. In the backward propagation, PGP is enabled and utilized to update the chosen prompts. (b) Process of prompt gradient projection. We sum input space and prompt space to obtain the sum space. Then with SVD, we attain the new orthogonal vectors from the sum space and update the projection matrix, which is described in Appendix C. Finally, we project the gradient by multiplying with the projection matrix.

concatenated with visual embedding sequences for prediction. During the backward propagation, we modify the prompt gradient by meeting the requirement of orthogonal condition from gradient projection methods. Figure 2(b) shows the process of prompt gradient projection. We use the element-wise sum to obtain the so-called sum space. With SVD in this space, we obtain the gradient projection matrix, and modify the gradient with this projection matrix to finish the PGP process.

## 3.1 PROMPT GRADIENT PROJECTION

From the perspective that the old inputs from previous tasks have the same outputs after learning a new task, we have the following proposition:

**proposition 1** *To better preserve old knowledge, the update of network would satisfy the following equation:*

$$f_\theta(p_{t+1}, x_t) = f_\theta(p_t, x_t), \tag{2}$$

where $x_t$ denotes the feature embeddings from old task $t$, $p_t$ and $p_{t+1}$ denote the prompts trained at task $t$ and $t + 1$, respectively. Proposition 1 depicts the mathematical, or ideal condition of anti-forgetting. In previous gradient projection works (Saha et al., 2021; Wang et al., 2021), it could be achieved by (i) limiting gradient direction to minimal interference with old knowledge; (ii) projecting gradient onto orthogonal space of old inputs. But as a byproduct, both require the task identifier, an additional prerequisite for inference.

In order to realize proposition 1 unlimitedly, we start from the implementation of prompt-based continual learning (PCL). In this framework, after the training of task $t + 1$, we concatenate the prompts $p_{t+1}$ and the embedding sequences $x_t$, *i.e.,* inputs from $t$-th task, along the embedding dimension: $Z_t^{t+1} = \begin{bmatrix} p_{t+1} \\ x_t \end{bmatrix}$. With the weights of $W_q$, $W_k$, $W_v$, PCL adopts the transformer architecture that allows us to obtain query ($Q_t^{t+1} = W_q Z_t^{t+1}$) and key ($K_t^{t+1} = W_k Z_t^{t+1}$). Thus the attention matrix (Dosovitskiy et al., 2020) is calculated as:

$$A_t^{t+1} = softmax(\frac{Q_t^{t+1} K_t^{t+1^T}}{\sqrt{(\frac{d}{h})}}). \tag{3}$$

Here, the denominator denotes a normalized factor and hence our focus turns to the numerator part $Q_t^{t+1} K_t^{t+1^T}$. It actually can be further expanded as $W_q Z_t^{t+1} Z_t^{t+1^T} W_k^T$. Notice that $W_q$ and $W_k$, the weights of visual encoder, are frozen and unchanged during training. The trainable parameters can be denoted as:

$$Z_t^{t+1} \cdot Z_t^{t+1^T} = \begin{bmatrix} p_{t+1} \\ x_t \end{bmatrix} \begin{bmatrix} p_{t+1}^T & x_t^T \end{bmatrix} = \begin{bmatrix} p_{t+1}p_{t+1}^T & p_{t+1}x_t^T \\ x_t p_{t+1}^T & x_t x_t^T \end{bmatrix}. \tag{4}$$

By contrast, the old embedding $Z_t^t$, is obtained through the concatenating prompts trained at task $t$ and embedding sequences $x_t$:

$$Z_t^t \cdot Z_t^{t^T} = \begin{bmatrix} p_t \\ x_t \end{bmatrix} \begin{bmatrix} p_t^T & x_t^T \end{bmatrix} = \begin{bmatrix} p_t p_t^T & p_t x_t^T \\ x_t p_t^T & x_t x_t^T \end{bmatrix}. \tag{5}$$

To achieve Eq.(2), *i.e.,* the condition of anti-forgetting, the new prompts require to be:

$$\begin{cases} p_{t+1}p_{t+1}^T = p_t p_t^T, \\ x_t p_{t+1}^T = x_t p_t^T, \\ p_{t+1}x_t^T = p_t x_t^T. \end{cases} \tag{6}$$

In Eq.(6), we divide $p_{t+1}$ into $p_t$ and $\Delta p$, where $\Delta p$ is the gradient of prompts when training task $t + 1$[2]. Therefore, for the first term, we extend $p_{t+1}p_{t+1}^T$ as:

$$p_{t+1}p_{t+1}^T = (p_t + \Delta p)(p_t + \Delta p)^T = p_t p_t^T + p_t \Delta p^T + \Delta p p_t^T + \Delta p \Delta p^T. \tag{7}$$

Here we ignore the high-order infinitesimal term of $\Delta p \Delta p^T$. Thus if $p_t \Delta p^T = 0$, the condition, *i.e.,* $p_{t+1}p_{t+1}^T = p_t p_t^T$ can be realized. In the same way, the second condition can be transformed to:

$$x_t p_{t+1}^T = x_t(p_t^T + \Delta p^T) = x_t p_t^T + x_t \Delta p^T = x_t p_t^T. \tag{8}$$

Eliminating $x_t p_t^T$ on both sides, we have $x_t \Delta p^T = 0$. Note that this condition also satisfies the third term in Eq.(6) because $x_t p_{t+1}^T$ is the transpose of $p_{t+1}x_t^T$. For prefix-tuning, this condition is also deduced, and included in Appendix E.

Therefore, our key observation is reached: restricting the gradient of prompts by the following equations can realize anti-forgetting:

$$\begin{cases} x_t \Delta p^T = 0, \\ p_t \Delta p^T = 0. \end{cases} \tag{9}$$

To solve this equation, we decompose $x_t$ with SVD: $x_t = U_t \Sigma_t V_t^T$. Here, $U_t$ and $V_t$ contain singular vectors corresponding to singular values in $\Sigma_t$, and diagonal matrix $\Sigma_t$ can be further divided as:

$$\Sigma_t = \begin{bmatrix} \Sigma_{t,1} & O \\ O & \Sigma_{t,0} \end{bmatrix}, \tag{10}$$

where $\Sigma_{t,1}$ denotes the non-zero elements of $\Sigma_t$ (non-zero singular values) and $\Sigma_{t,0}$ denotes the near-zero elements of $\Sigma_t$ (Deisenroth et al., 2020). Correspondingly, $V_t$ can be divided into two parts along the column dimension: $V_t = [V_{t,1}, V_{t,0}]$. Thus, we have:

$$x_t[V_{t,1}, V_{t,0}] = U_t \begin{bmatrix} \Sigma_{t,1} & O \\ O & \Sigma_{t,0} \end{bmatrix}. \tag{11}$$

As a result, we obtain the following equation:

$$x_t V_{t,0} = U_t \begin{bmatrix} O \\ \Sigma_{t,0} \end{bmatrix} \approx O. \tag{12}$$

Let $\Delta p = \Delta p V_{t,0} V_{t,0}^T$, we can get:

$$x_t \Delta p^T = x_t(\Delta p V_{t,0} V_{t,0}^T)^T = x_t V_{t,0} V_{t,0}^T \Delta p^T = O. \tag{13}$$

Eq.(13) allows us to successfully meet the first requirement in Eq.(9), by taking $V_{t,0}$ as the gradient projection matrix. We also have a similar conclusion for the second requirement in Eq.(9). In fact, to simplify the implementation process of Eq.(9), we combine $p_t$ and $x_t$ with element-wise sum:

$$s_t = x_t + p_t. \tag{14}$$

Thus we conduct SVD on $s_t$ and therefore the obtained projection matrix $V_{t,0}$ can realize $s_t \Delta p^T = 0$, which equals to $x_t \Delta p^T = 0$ and $p_t \Delta p^T = 0$. In the training stage, we update the gradient with a projection of $\Delta p' = \Delta p V_{t,0} V_{t,0}^T$.

---

[2] Here we omit the factor of learning rating since this simplification wouldn't influence our conclusion.

## 3.2 GRADIENT PROJECTION FOR PROMPT POOL

For prompt pool, there is another learnable parameter: key. Prompt-based continual learning often deploys a query-key pair for seeking the matched prompts. In this case, the old knowledge would also be interfered with, as the update on key would influence this matching process. Fortunately, the gradient projection method would be generalized well in such case.

First of all, let us recall the pipeline of PCL. To choose the relevant prompts, we first calculate cosine similarity between query feature and key:

$$\phi(q, k) = \frac{q^T k}{||q|| \cdot ||k||}, \tag{15}$$

where $q$, $k$ represent query feature and key respectively. To achieve anti-forgetting in the scenario of prompt pool, we have the following proposition:

**proposition 2** *Old knowledge could be preserved, if the following equation holds:*

$$q_t^T k_{t+1} = q_t^T k_t. \tag{16}$$

For further illustration, we expand $k_{t+1}$ with $k_t$ and $\Delta k$, where $\Delta k$ is the gradient change from task $t$ to task $t + 1$, and have:

$$q_t^T k_{t+1} = q_t^T (k_t + \Delta k) = q_t^T k_t + q_t^T \Delta k. \tag{17}$$

The above formulations suggest $q_t^T \Delta k = 0$, which is similar to Eq.(9). In fact, we notice that, $q_t^T \Delta k = 0$ is slightly different from Eq.(9), that it uses the transposition form. Therefore in our implementation, when sampling the orthogonal space of query features, we need first to transpose this feature matrix, like $q_t^T = V_t \Sigma_t U_t^T$.

## 3.3 BALANCE BETWEEN STABILITY AND PLASTICITY

We consider singular value decomposition (SVD) of $s_t$ as $s_t = \hat{U}_t \hat{\Sigma}_t \hat{V}_t^T$. $\hat{V}_t$ consists of singular vectors decomposed of $s_t$. Here, we define a threshold, $\epsilon \in [0, 1]$ to split $\hat{V}_t$ into two parts $\hat{V}_t = [\hat{V}_t^1, \hat{V}_t^2]$, where $\hat{V}_t^1 = \hat{V}_t[:, 1 : \epsilon n]$, $\hat{V}_t^2 = \hat{V}_t[:, \epsilon n : n]$ and $n$ is the column size of $\hat{V}_t$. We project the gradient $g_{t+1}$ as:

$$g_{t+1}{}' = g_{t+1} \hat{V}_t^2 \hat{V}_t^{2T}. \tag{18}$$

Therefore, there are three situations for $\hat{V}_t^2$. Firstly, if $\hat{V}_t^2 = 0$, we have:

$$g_{t+1}{}' = O. \tag{19}$$

In this situation, all trainable parameters are frozen and the old knowledge will be preserved completely. Secondly, if $\hat{V}_t^2 = V_{t,0}$, we have[3]:

$$g_{t+1}{}' = g_{t+1} V_{t,0} V_{t,0}{}^T. \tag{20}$$

In this situation, we update prompts by projecting gradient onto the orthogonal space of old inputs. Hence, samples from old tasks can have the same outputs for the new model. The old knowledge will be preserved well and the model can also learn some new knowledge with updating. Thirdly, if $\hat{V}_t^2 = \hat{V}_t$, we have:

$$g_{t+1}{}' = g_{t+1} \hat{V}_t^2 \hat{V}_t^{2T} = g_{t+1} \hat{V}_t \hat{V}_t^T = g_{t+1}. \tag{21}$$

In this situation, the parameters are updated normally without projection. In our implementation, we rearrange $\hat{V}_t$ sorted according to the corresponding singular values. We use $\epsilon$ to control this balance. More detailed discussion could be seen in Appendix B.

---

[3]Here $V_{t,0}$ is obtained by decomposition of $s_t$ mentioned above.

Table 1: Main results of class incremental learning in terms of accuracy and forgetting on 10-Split-CIFAR100, 20-Split-CIFAR100, and 10-Split-ImageNet-R. Exemplar means the total buffer size for rehearsal methods. For detailed metrics information please refer to Appendix F.

| Method | Exemplar | 10-Split-CIFAR100 | | 20-Split-CIFAR100 | | 10-Split-ImageNet-R | |
|---|---|---|---|---|---|---|---|
| | | ACC($\uparrow$) | Forgetting($\downarrow$) | ACC($\uparrow$) | Forgetting($\downarrow$) | ACC($\uparrow$) | Forgetting($\downarrow$) |
| BiC | 5000 | 81.42 | 17.31 | 73.02 | 6.23 | 64.63 | 22.25 |
| DER++ | 5000 | 83.94 | 14.55 | - | - | 66.73 | 20.67 |
| ICaRL | 5000 | 66.00 | 5.33 | 78.02 | 5.80 | - | - |
| DER+MCG | 2000 | 67.62 | 14.64 | 65.84 | 13.72 | - | - |
| BiC | 1000 | 66.11 | 35.24 | 63.12 | 21.89 | 52.14 | 36.70 |
| DER++ | 1000 | 61.06 | 39.87 | - | - | 55.47 | 34.64 |
| ICaRL | 1000 | 61.25 | 14.19 | 71.32 | 15.98 | - | - |
| FT | ✗ | 33.61 | 86.87 | 33.52 | 53.69 | 28.87 | 63.80 |
| EWC | ✗ | 47.01 | 33.27 | 36.73 | 35.19 | 35.00 | 56.16 |
| LWF | ✗ | 60.69 | 27.77 | 39.12 | 57.91 | 38.54 | 52.37 |
| L2P | ✗ | 83.77 | 6.63 | 81.29 | 8.96 | 60.44 | 9.00 |
| **L2P-PGP(Ours)** | ✗ | **84.34** | **5.59** | **82.00** | **8.39** | **61.40** | **8.03** |
| DualPrompt | ✗ | 86.50 | 5.77 | 82.98 | 8.20 | 68.13 | 4.68 |
| **DualPrompt-PGP(Ours)** | ✗ | **86.92** | **5.35** | **83.74** | **7.91** | **69.34** | **4.53** |
| Upper-Bound | - | 90.85 | - | 90.85 | - | 79.13 | - |

# 4 EXPERIMENTAL SETUP

**Datasets:** We evaluate our method on **1) 10/20-Split-CIFAR100** (Krizhevsky et al., 2009), constructed by splitting the 100 classes into 10 tasks/20 tasks. **2) 10-Split-TinyImageNet** (Abai & Rajmalwar, 2019), constructed by splitting the 200 classes into 10 tasks. **3) 10-Split-ImageNet-R** (Hendrycks et al., 2021), constructed by splitting the 200 classes into 10 tasks.

**Implementation:** We use L2P (Wang et al., 2022c), DualPrompt (Wang et al., 2022b), and CLIP (Radford et al., 2021) as our baselines, with prompt gradient projection for updating. We follow their original settings, and the only difference is we train DualPrompt with extra 15 epochs on CIFAR100 suggested by (Khan et al., 2023). Detailed experiment information could be seen in Appendix G.

**Competitors:** We compare our results with representative SOTA CIL methods including ICaRL (Rebuffi et al., 2017), BiC (Wu et al., 2019), DER++ (Buzzega et al., 2020), LWF (Li & Hoiem, 2017), EWC (Kirkpatrick et al., 2017), DER+MCG (Cai et al., 2023). We adopt average accuracy (simplified as accuracy/ACC) and forgetting (simplified as FOR) as our validation metrics (Wang et al., 2022b). Results and comparisons of task incremental learning can be found in Appendix J.

# 5 RESULTS AND DISCUSSION

**Class Incremental Setting**[4]: We compare our method with state-of-the-art CIL approaches, and the main results are shown in Table 1. We observe that DualPrompt with PGP obtains the best results and achieves a new SOTA. When comparing DualPrompt with DualPrompt-PGP, it appears that PGP brings a decent improvement in anti-forgetting. On 10-Split-CIFAR100, PGP improves DualPrompt by 0.42% on forgetting and 0.42% on accuracy. Similarly, on 20-Split-CIFAR100, PGP improves DualPrompt by 0.29% on forgetting and 0.76% on accuracy, and on 10-Split-ImageNet-R, PGP improves DualPrompt by 0.15% on forgetting and 1.21% on accuracy.

For L2P, PGP also brings evident performance improvements. On 10-Split-CIFAR100, PGP obtains an improvement of 1.04% on forgetting and 0.43% on accuracy. On 10-Split-ImageNet-R, our method also obtains an improvement of 0.97% on forgetting and 0.96% on accuracy.

**Analysis of Training Time and Memory Space**: We present the comparison between L2P-PGP and L2P-R (L2P with rehearsal exemplar) in terms of training time and memory cost in Table 2. For a fair comparison, we maintain complete consistency in experimental settings such as batch size, training epoch, and prompt length et al.

---

[4]Experiment results of CLIP model please refer to Appendix K.

It is worth noting that our method doesn't require any exemplar for rehearsal and therefore not only uses less memory space, but avoids the privacy leaking problem (Shokri & Shmatikov, 2015) as well. At the same time, our approach has a lower forgetting and shorter training time.

Table 2: Comparison of ACC, forgetting, memory, and training time between L2P-PGP and L2P-R.

| Method | Exemplar | ACC(↑) | Forgetting(↓) | Memory | Training Time |
|---|---|---|---|---|---|
| L2P-R | 1000 | 84.21 | 7.72 | 1.12 GB | 0.787h |
| **L2P-PGP** | ✗ | **84.26** | **5.64** | **≤1 MB** | **0.756 h** |

**Online Class Incremental Setting**: online class incremental learning is a challenging class incremental task that only allows training each task for one epoch. We compare PGP and its baseline on this setting shown in Table 3.

Table 3: Main results of online class incremental learning in terms of accuracy and forgetting. The comparison is made between our approach and the corresponding baselines.

| Method | 10-Split-CIFAR100 | | 20-Split-CIFAR100 | | 10-Split-TinyImageNet | |
|---|---|---|---|---|---|---|
| | ACC(↑) | Forgetting(↓) | ACC(↑) | Forgetting(↓) | ACC(↑) | Forgetting(↓) |
| L2P | 79.99 | 8.19 | 77.63 | 11.33 | 78.69 | 5.83 |
| **L2P-PGP** | **80.29** | **7.73** | **78.34** | **9.33** | **79.47** | **5.19** |
| DualPrompt | 80.93 | 5.51 | 79.02 | 6.89 | 82.20 | 3.62 |
| **DualPrompt-PGP** | **81.02** | **5.41** | **79.41** | **6.75** | **82.57** | **3.57** |

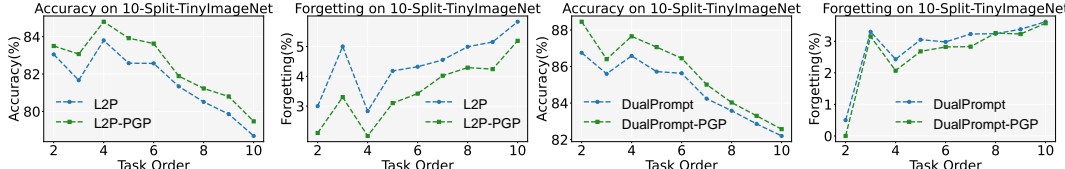

Figure 3: Task-by-task performance changing curves in terms of accuracy and forgetting under online class incremental setting.

On 10/20-Split-CIFAR100 and 10-Split-TinyImageNet, it is observed that PGP is able to improve accuracy and reduce forgetting for both L2P and DualPrompt. On 10-Split-CIFAR100 and 20-Split-CIFAR100, we discover that our method can improve L2P by 0.30% and 0.71% on accuracy respectively, while reducing 0.46% and 2.00% on forgetting. On 10-Split-TinyImageNet, we also find that our method improves L2P by 0.78% on accuracy and 0.64% on forgetting. Similar to L2P, for DualPrompt, we take 20-Split-CIFAR100 dataset as an example, PGP brings the method improvement of 0.39% on accuracy and 0.14% on forgetting.

Figure 3 shows the curves of accuracy and forgetting with the task number increasing on 10-Split-TinyImageNet. We observe that on all tasks, accuracy of our method is always higher than baseline, and forgetting is always lower than baseline. These phenomena demonstrate that our method has advantages over baseline with fewer training epochs.

# 6  ABLATION STUDY

**Impact of Projection Settings:** As shown in Table 4 and Figure 4, we evaluate the results by performing gradient projection on the gradient of prompt (l2p-p), key (l2p-k) and both of them (l2p-pk), respectively. Original L2P is named "l2p-o". Table 4 quantitatively indicates that l2p-pk has the best anti-forgetting performance since it has the strictest constraint. Concretely, compared with l2p-o, l2p-p, l2p-k and l2p-pk decreases the forgetting by 0.99%, 0.76%, and 1.04%, while increasing the accuracy by 0.49%, 0.30% and 0.57% on 10-Split-CIFAR100. On 10-Split-TinyImageNet, l2p-p, l2p-k, and l2p-pk decrease the forgetting by 0.42%, 0.34%, and 0.78%, while increasing the accuracy by 0.65%, 0.37%, and 0.73% in comparison with l2p-o.

We have observed that l2p-p performs better than l2p-k in both terms of accuracy and forgetting. The reason might be the prompt directly participates in the image encoding process. Figure 4 also

Table 4: Ablation study of various gradient projection manners. l2p-o, l2p-p, l2p-k, and l2p-pk denote the original L2P, gradient projection on prompt, key, and both prompt and key, respectively.

| Method | 10-Split-CIFAR100 | | 10-Split-TinyImageNet | |
|---|---|---|---|---|
| | Forgetting(↓) | ACC(↑) | Forgetting(↓) | ACC(↑) |
| l2p-o | 6.63 | 83.77 | 5.68 | 81.92 |
| l2p-p | 5.64 | 84.26 | 5.26 | 82.57 |
| l2p-k | 5.87 | 84.07 | 5.34 | 82.29 |
| l2p-pk | 5.59 | 84.34 | 4.90 | 82.65 |

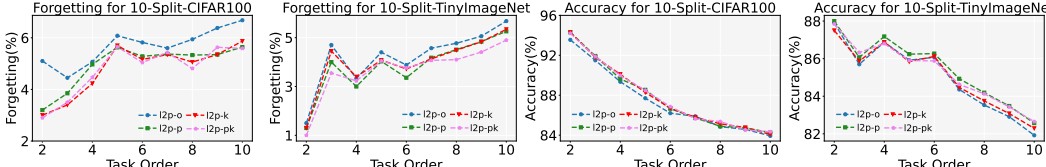

Figure 4: Task-by-task performance changing curves in terms of accuracy and forgetting under various gradient projection manners.

shows the changes of forgetting values with the task number increasing, where the line of l2p-p, l2p-k, and l2p-pk is always below l2p-o.

**Impact of Distinct Threshold:** We study the hyperparameter sensitivity by setting $\epsilon$ with values in [0.60, 0.70, 0.80]. and conduct experiments on 10-Split-CIFAR100 and 10-Split-TinyImageNet datasets respectively, shown in Figure 5.

In Figure 5, we can clearly find that with $\epsilon$ increasing, forgetting shows a clear degradation, indicating that the ability of anti-forgetting (stability) becomes stronger, while the accuracy (plasticity) shows a trend of decline. These two phenomena perfectly illustrate that if $\hat{V}_t^1$ has fewer columns (low $\epsilon$), the model would have better plasticity but worse stability, and if $\hat{V}_t^2$ has fewer columns (high $\epsilon$), the model would have better stability but worse plasticity. The above conclusions also have been reported in GPM (Saha et al., 2021) and Adam-NSCL (Wang et al., 2021). Thus, gradient projection is also a trade-off between plasticity and stability when meeting prompt-based continual learning.

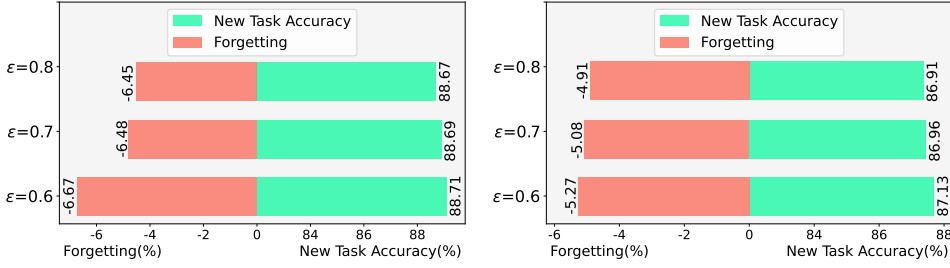

Figure 5: Performance histograms in terms of forgetting and new task accuracy by varying $\epsilon$.

## 7 CONCLUSION

In this paper, we propose prompt gradient projection, which deduces the gradient condition for prompt to reduce forgetting. Then the gradient projection matrix is obtained by conducting SVD on a sum space. Finally, we discuss how to balance plasticity and stability from the perspective of gradient projection. We validate our approach in benchmark datasets under various incremental settings and demonstrate the effectiveness of our approach. This paper is an initial attempt towards combining prompt-tuning and gradient projection. We hope our work would inspire the focus on the anti-forgetting mechanism of prompt-based continual learning and can be extended to more parameter-efficient paradigms *i.e.,* adapter-tuning and LoRA-tuning, and large models.

## 8 ACKNOWLEDGMENT

This work is supported by the National Key Research and Development Program of China (2021ZD0111000), Science and Technology Commission (No.21511100700), National Natural Science Foundation of China (No.62222602, No.62106075, No.62176092, No.62302167, No.62176224, No.61972157 No.U23A20343, No.72192821), Natural Science Foundation of Shanghai (23ZR1420400), Shanghai Sailing Program (23YF1410500) and CAAI-Huawei Mind-Spore Open Fund.

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

## A    APPENDIX

## B    PROOF OF INSIGHT WITH BALANCING STABILITY AND PLASTICITY

**Lemma 1** *Singular value decomposition (SVD): For matrix $A_{m,n}$, we can factorize it into three matrices and obtain matrix $U_{m,l}$, $\Sigma_{l,l}$, and $V_{l,n}$, where $U_{m,l}$ and $V_{l,n}$ are orthogonal matrices, $\Sigma_{l,l}$ contains the sorted singular value along its main diagonal:*

$$A_{m,n} = U_{m,l}\Sigma_{l,l}V_{l,n}{}^T. \tag{22}$$

**Theorem 1** *For any embedding sequences $x_t$, embedded from samples of task $t$, using singular value decomposition (SVD) in lemma 1, we can obtain matrix $U_t$, $\Sigma_t$, and $V_t$. Then, we randomly split $V_t$ along the column dimension into two parts: $[V_t^1, V_t^2]$. Because $V_t$ is an orthogonal matrix, we can have:*

$$V_t^l V_t^{l\,T} = [V_t^1 V_t^2]\begin{bmatrix} V_t^{1T} \\ V_t^{2T} \end{bmatrix} = V_t^1 V_t^{1T} + V_t^2 V_t^{2T} = I. \tag{23}$$

**On Plasticity:** For any matrix $V$ and gradient $g_{t+1}$ when learning task $t + 1$, we have the gradient after projection on $V$ as $g_{t+1}^{'}$, and calculate the cosine similarity between $g_{t+1}$ and $g_{t+1}^{'}$:

$$
\begin{aligned}
< g_{t+1}^{'}, g_{t+1} > &=< g_{t+1}VV^T, g_{t+1} > \\
&= \text{vec}(g_{t+1}VV^T I)^T \text{vec}(g_{t+1}) \\
&= \text{vec}(g_{t+1}V)^T (I)(I \times V)\text{vec}(g_{t+1}) \\
&= \text{vec}(g_{t+1}V)^T \text{vec}(g_{t+1}V) \\
&\geq 0,
\end{aligned}
\tag{24}
$$

which means that whatever $V$ is, as if $g_{t+1}$ and $V$ both are not equal to zero, after projection on $V$, the new gradient is always positive. Thus, the model is always learning new knowledge. The only distinction is the gradient direction, which can be measured by calculation of cosine similarity between gradient before projection and after projection. We set gradient project matrix as $V_t^2$ and have:

$$
\begin{aligned}
< g_{t+1}^{'}, g_{t+1} > &=< g_{t+1}V_t^2 V_t^{2T}, g_{t+1} > \\
&=< g_{t+1}(I - V_t^1 V_t^{1T}), g_{t+1} > \\
&=< g_{t+1} - g_{t+1}V_t^1 V_t^{1T}, g_{t+1} > \\
&=< g_{t+1}, g_{t+1} > - < g_{t+1}V_t^1 V_t^{1T}, g_{t+1} > \\
&\leq< g_{t+1}, g_{t+1} > .
\end{aligned}
\tag{25}
$$

From the inequality, if projection matrix $V_t^2$ is not an identity matrix, which means that matrix $V_t^1$ is not a zero matrix, the direction of gradient after projection always has an angle with the direction of the original gradient, incurring that decrease of loss function becomes slow.

For further research on the relationship between matrix $V_t^1$ and decreased speed of loss function, we mainly focus on two situations for value of $V_t^1$:

(1).$V_t^1 = 0$, we have

$$< g_{t+1}^{'}, g_{t+1} >=< g_{t+1}, g_{t+1} > . \tag{26}$$

In this situation, it equals that we do not operate on the gradient, and parameters are updated normally. When $V_t^1 = 0$, it has $V_t^2 = V_t$. In fact, the same phenomenon has been shown in the previous situation that $V_t^2 = V_t$.

(2).$V_t^1 = V_t$, we have:

$$
\begin{aligned}
< g_{t+1}^{'}, g_{t+1} > &=< g_{t+1}, g_{t+1} > - < g_{t+1}V_t V_t^T, g_{t+1} > \\
&=< g_{t+1}, g_{t+1} > - < g_{t+1}, g_{t+1} > \\
&= O.
\end{aligned}
\tag{27}
$$

In this situation, it equals that we freeze the network update process, and trainable parameters are stable and not changed. Thus, the network will not adopt any other new knowledge. When $V_t^1 = V_t$, it has $V_t^2 = 0$. In fact, the same phenomenon has been shown in the previous situation that $V_t^2 = 0$.

In conclusion, with $V_t^1$ changing from 0 to $V_t$, the decreased speed of the loss function becomes more and more slow, leading to worse plasticity. However, under this trend, $V_t^2$ is changing from $V_t$ to 0, giving the anti-forgetting more and more strength. We can recognize that the essence of the gradient projection method is a kind of trade-off strategy between plasticity and stability. However, different from other dilemmas, it has an optimal solution, which is projecting gradient in the direction orthogonal to the subspace spanned by the old inputs, which can not only own the best ability of anti-forgetting, but also have minimal damage to plasticity.

## C  METHOD OF UPDATING PROJECTION MATRIX

We update our projection matrix $V_{t,0}$ like GPM (Saha et al., 2021), which is detailed described as follows. Assume that we have sampled embedding sequences from current task samples $x_t$ and trained prompts $p_t$. Here, $t$ is the task identifier. We utilize Principal Component Analysis (PCA) to compress and align the dimensions of $x_t$ and $p_t$. Then, we element-wise add $x_t$ and $p_t$ to obtain $s_t$. Besides that, we set a threshold $\epsilon$.

For task #1 training, we perform SVD on $s_1$ as $s_1 = U_1 \Sigma_1 V_1^T$. We collect the minimum former $l$ columns of $V_1$ as matrix $L = [v_{11}, v_{12}, ..., v_{1l}]$ according to the following criteria:

$$||s_{1l}||_F^2 \geq \epsilon ||s_1||_F^2. \tag{28}$$

Here, $||.||_F$ is the Frobenius norm of the matrix and $\epsilon$ ($0 < \epsilon \leq 1$) is the threshold hyperparameter. $V_{1,0}$ can be obtained by $V_{1,0} V_{1,0}^T = I - LL^T$.

For task #2 training, before performing SVD and subsequent former-rank approximation, we eliminate the common directions in $s_2$ which are already present in $L$, so that newly added column vectors are unique and orthogonal to the existing column vectors. Thus, we perform the step $\hat{s}_2 = s_2 - LL^T s_2$. Afterward, SVD is performed on $\hat{s}_2 (= \hat{U}_2 \hat{\Sigma}_2 \hat{V}_2^T)$ and former $m$ new columns of $\hat{V}_2$ are chosen with minimum value of $m$ satisfying the following criteria:

$$||LL^T s_2||_F^2 + ||\hat{s}_{2m}||_F^2 \geq \epsilon ||s_2||_F^2. \tag{29}$$

Here, $L$ is updated by adding new column vectors as $[v_{11}, v_{12}, ..., v_{1l}, \hat{v}_{21}, \hat{v}_{22}, ..., \hat{v}_{2m}]$. Then, we can update $V_{1,0}$ to $V_{2,0}$ according to $V_{2,0} V_{2,0}^T = I - LL^T$. Once the update is complete we move on to the next task and repeat the same procedure as in task #2.

## D  COMPUTATION COST FOR MAINTAINING THE ORTHOGONALITY OF THE TASK SUBSPACES

In this section, we will discuss the added computation cost for maintaining the orthogonality of the task subspaces under the following situations.

### D.1  LARGER MODELS

If we change the backbone from a smaller one to a larger one, it could have different results of added computation cost for distinct tuning paradigms. i) For prompt-tuning, because we only prepend the prompt into the first transformer layer, the added computation could be omitted. ii) For prefix-tuning, larger models usually mean more network layers or wider input dimensions, and we need to expand the prefix-inserted layer or prefix width, which is the origination of the added computation cost. For expanding the prefix-inserted layer, each layer can have a nearly similar computation cost if the number of samples is the same. Thus, we can conclude that the added computation cost can be modeled as an approximate linear function with the layer numbers of the backbone. Similarly, the same conclusion can also be drawn from expanding the prefix width.

## D.2 INCREASED NUMBER OF TASKS

Observing the training processes of multi-datasets, we can empirically summarize that in each task, the number of newly added column vectors of the projection matrix is constant in a certain range. As the added computation cost is mainly focused on i) calculation of the projection matrix and ii) multiplication between the projection matrix and its transpose, we can see that although it could not appear exponential explosion, it is still a potential risk in our method with the increased number of tasks.

## E GRADIENT PROJECTION BASED ON PREFIX-TUNING PARADIGM

In this section, we prove that the gradient projection method can be utilized in prefix-tuning with mathematical deduction.

Distinct from prompt-tuning paradigm, prefix-tuning only prepends prefixes in key vector and value vector, without query vector of prepended transformer layer. Additionally, different from prompt usually only prepended in the first transformer layer, prefix can be prepended in any transformer layers. These advantages help models based on prefix-tuning own a better performance than those based on prompt-tuning both in natural language processing and computer vision.

For baseline based on prefix-tuning, if we want to preserve old knowledge, we need to realize:

$$f_\theta(p_{t,l}, x_{t,l}) = f_\theta(p_{t+1,l}, x_{t,l}). \tag{30}$$

$f_\theta$ refers to ViT model, $x_{t,l}$ denotes inputs at task $t$ in layer $l$, $p_{t,l}$ and $p_{t+1,l}$ represents the prefixes trained at task $t$ and prepended in layer $l$ and the prefixes trained at task $t+1$ and prepended in layer $l$ respectively.

Assuming that a set of prefixes have been trained at task $t+1$, and we input samples from task $t$. Now, we prepend prefix in key vector, and have:

$$Q_{t,l} = W_{q,l} x_{t,l}, \tag{31}$$

$$K_{t,l} = \begin{bmatrix} p_{t+1,l} \\ W_{k,l} x_{t,l} \end{bmatrix}, \tag{32}$$

where, $W_{q,l}$ and $W_{k,l}$ are weights of ViT, frozen and unchanged. With Eq.(3), we have the results that $t$-th task samples on $t+1$-th model. We mainly focus on the part:

$$Q_{t,l} K_{t,l}^T = W_{q,l} x_{t,l} \begin{bmatrix} p_{t+1,l}^T & (W_{k,l} x_{t,l})^T \end{bmatrix} = \begin{bmatrix} W_{q,l} x_{t,l} p_{t+1,l}^T & W_{q,l} x_{t,l} x_{t,l}^T W_{k,l}^T \end{bmatrix}. \tag{33}$$

As stable item $W_{q,l} x_{t,l} x_{t,l}^T W_{k,l}^T$, we only focus on the item $W_{q,l} x_{t,l} p_{t+1,l}^T$. Changing $p_{t+1,l}^T$ with $p_{t,l}^T$, we can obtain the results that $t$-th task samples on $t$-th model. Because our aim is making $W_{q,l} x_{t,l} p_{t+1,l}^T$ equal to $W_{q,l} x_{t,l} p_{t,l}^T$, considering that $W_{q,l}$ is frozen, our final aim can be simplified as:

$$x_{t,l} p_{t+1,l}^T = x_{t,l} p_{t,l}^T, \tag{34}$$

which has the same form as Eq.(8), meaning that we can also achieve Eq.(34) by the gradient projection method. Thus, we can draw the conclusion that the gradient projection method could also help models based on prefix-tuning to resist forgetting.

# F    METRICS

**Two metrics:** Average Accuracy (simplified as accuracy/ACC) and Forgetting (simplified as FOR) are used to evaluate the performance. We use average accuracy metric, for averaging the classification accuracy of all classes. We adopt forgetting metric to indicate the average loss of accuracy of past tasks after learning a new task. Formally, average accuracy and forgetting are defined as:

$$\text{Average Accuracy} = \frac{1}{T} \sum_{i=1}^{T} A_{T,i}, \tag{35}$$

$$\text{Forgetting} = \frac{1}{T-1} \sum_{i=1}^{T-1} A_{T,i} - \max(A_{j,i})_{j \in [i, T-1]}, \tag{36}$$

where $T$ is the number of tasks, $A_{T,i}$ is the accuracy of $i$-th task samples on the $T$-th model, and $A_{j,i}$ is the accuracy of $i$-th task samples on the $j$-th model.

# G    EXPERIMENTAL DETAILS

Consistent with previous works (Wang et al., 2022c;b; Smith et al., 2023), we use ViT B/16 (Dosovitskiy et al., 2020) pre-trained on ImageNet-21K as our image encoder, which is kept frozen during training. We train and test on one A6000-48GB GPU for baselines and our method. We set the Adam optimizer with $\beta_1 = 0.9$ and $\beta_2 = 0.999$.

For hyperparameters, in L2P-PGP, we set $\epsilon = 0.50$ for extraction of prompt gradient projection matrix and $\epsilon = 0.97$ for key gradient projection matrix. While in DualPrompt-PGP, we set $\epsilon = 0.50$ for extraction of prompt gradient projection matrix. To accelerate the speed of gradient projection matrix extraction and reduce the training space, we add PCA into our process, which can be used to compress the sampled feature space.

In comparison with L2P and L2P-PGP, for 10/20-Split-CIFAR100, and 10-Split-TinyImageNet, we both train the network for 5 epochs with batch size of 16 and prompt length is set at 5, while we both set epochs as 50, batch size as 16, and prompt length as 30 for 10-Split-ImageNet-R.

In comparison with DualPrompt and DualPrompt-PGP, for 10/20-Split-CIFAR100, we train the network for 20 epochs with batch size of 24, and expert prompt length is set at 5. While we both set epochs as 5, batch size as 24, and expert prompt length as 5 for 10-Split-TinyImageNet, epochs as 50 and batch size as 24 for 10-Split-ImageNet-R with expert prompt length at 20. Besides that, in all benchmark datasets, the general prompt length is set at 5 and the prompt-inserted locations are kept the same.

For CLIP-PGP, the experimental setting is that, on the vision side, we only set a single trainable image prompt shared by all tasks. As for the text side, we follow the operation as (Zhou et al., 2022), we set trainable text prompt for each class, which is only trained at the corresponding task. In comparison with CLIP and CLIP-PGP, we both set the image prompt length as 5, epochs as 5, and batch size as 32 for 10-Split-CIFAR100. Specifically in CLIP-PGP, we set $\epsilon = 0.90$ for extraction of image prompt gradient projection matrix.

# H    RESULT TABLE WITH THE STANDARD DEVIATION VALUES

We conduct 3 runs of our method and competitors, additional results with the standard deviation values on different datasets are shown in Table 5

# I    COMPARISON WITH BASELINES AND UPPER-BOUND

We compare the performance of prompt-based methods with and without PGP in Table 6. To be consistent with previous works (Wang et al., 2022c), we report the difference between accuracy performance of the Upper-Bound and the model as a metric. We observe that PGP again sets a new SOTA in this setting. As we compare the Diff performance of DualPrompt and L2P with and without PGP, we again notice an obvious improvement.

Table 5: Class incremental learning on different datasets along with the standard deviation values.

| Method | Exemplar | 10-Split-CIFAR100 | | 20-Split-CIFAR100 | | 10-Split-ImageNet-R | |
|---|---|---|---|---|---|---|---|
| | | ACC(↑) | Forgetting(↓) | ACC(↑) | Forgetting(↓) | ACC(↑) | Forgetting(↓) |
| BiC | 5000 | 81.42±0.85 | 17.31±1.02 | 73.02±0.93 | 6.23±1.17 | 64.63±1.27 | 22.25±1.73 |
| DER++ | 5000 | 83.94±0.34 | 14.55±0.73 | - | - | 66.73±0.87 | 20.67±1.24 |
| ICaRL | 5000 | 66.00±0.66 | 5.33±0.94 | 78.02±0.71 | 5.80±1.02 | - | - |
| DER+MCG | 2000 | 67.62±0.04 | 14.64±0.53 | 65.84±0.18 | 13.72±1.28 | - | - |
| BiC | 1000 | 66.11±1.76 | 35.24±1.64 | 63.12±2.35 | 21.89±1.93 | 52.14±1.08 | 36.70±1.05 |
| DER++ | 1000 | 61.06±0.87 | 39.87±0.99 | - | - | 55.47±1.31 | 34.64±1.50 |
| ICaRL | 1000 | 61.25±0.63 | 14.19±1.14 | 71.32±0.86 | 15.98±1.35 | - | - |
| FT | ✗ | 33.61±0.85 | 86.87±0.20 | 33.52±0.94 | 53.69±0.52 | 28.87±1.36 | 63.80±1.50 |
| EWC | ✗ | 47.01±0.29 | 33.27±1.17 | 36.73±0.57 | 35.19±1.98 | 35.00±0.43 | 56.16±0.88 |
| LWF | ✗ | 60.69±0.63 | 27.77±2.17 | 39.12±0.87 | 57.91±3.06 | 38.54±1.23 | 52.37±0.64 |
| L2P | ✗ | 83.77±0.16 | 6.63±0.05 | 81.29±0.43 | 8.96±0.38 | 60.44±0.41 | 9.00±0.86 |
| **L2P-PGP(Ours)** | ✗ | **84.34±0.08** | **5.59±0.05** | **82.00±0.56** | **8.39±0.62** | **61.40±0.34** | **8.03±0.03** |
| DualPrompt | ✗ | 86.50±0.45 | 5.77±0.02 | 82.98±0.47 | 8.20±0.08 | 68.13±0.10 | 4.68±0.19 |
| **DualPrompt-PGP(Ours)** | ✗ | **86.92±0.05** | **5.35±0.19** | **83.74±0.01** | **7.91±0.15** | **69.34±0.05** | **4.53±0.04** |
| Upper-Bound | - | 90.85±0.12 | - | 90.85±0.12 | - | 79.13±0.18 | - |

Table 6: Comparison with baselines in terms of differences between accuracy performance of the Upper-Bound and the model. The Upper-Bound denotes the model performance when trained with access to all tasks at the same time. we use **Diff = Upper-Bound ACC - Method ACC.**

| Method | 10-Split-CIFAR100 | | 20-Split-CIFAR100 | | 10-Split-ImageNet-R | |
|---|---|---|---|---|---|---|
| | ACC(↑) | Diff(↓) | ACC(↑) | Diff(↓) | ACC(↑) | Diff(↓) |
| Upper-Bound | 90.85 | - | 90.85 | - | 79.13 | - |
| L2P | 83.77 | 7.08 | 81.29 | 9.56 | 60.44 | 18.69 |
| **L2P-PGP** | **84.34** | **6.51** | **82.00** | **8.85** | **61.40** | **17.73** |
| DualPrompt | 86.50 | 4.35 | 82.98 | 7.87 | 68.13 | 11.00 |
| **DualPrompt-PGP** | **86.92** | **3.93** | **83.74** | **7.11** | **69.34** | **9.79** |

## J  TASK INCREMENTAL SETTING

We compare L2P-PGP with L2P and representative SOTA competitors: EWC (Kirkpatrick et al., 2017), LWF (Li & Hoiem, 2017), A-GEM (Chaudhry et al., 2018), OWM (Zeng et al., 2019), Adam-NSCL (Wang et al., 2021), Connector (Lin et al., 2022a), results as shown in Table 7.

Both on 10-Split-CIFAR100 and 20-Split-CIFAR100 datasets, although L2P has already achieved higher accuracy and lower forgetting compared with other CNN methods, our method further improves its accuracy and reduces its forgetting with the aid of prompt gradient projection and L2P-PGP achieves new SOTA performance. On 10-Split-CIFAR100 dataset, PGP improves L2P by 0.10 on accuracy, 0.05 on forgetting, and on 20-Split-CIFAR100, PGP improves L2P by 0.11 on accuracy, 0.11 on forgetting.

Table 7: Task incremental learning results on different datasets.

| Method | 10-Split-CIFAR100 | | 20-Split-CIFAR100 | |
|---|---|---|---|---|
| | ACC(↑) | Forgetting(↓) | ACC(↑) | Forgetting(↓) |
| EWC | 70.77 | 2.83 | 71.66 | 3.72 |
| LWF | 70.70 | 6.27 | 74.38 | 9.11 |
| A-GEM | 49.57 | 1.13 | 61.91 | 6.88 |
| OWM | 68.89 | 1.88 | 68.47 | 3.37 |
| Adam-NSCL | 73.77 | 1.60 | 75.95 | 3.66 |
| Connector | 79.79 | 0.92 | 80.80 | 5.00 |
| L2P | 97.43 | 0.22 | 98.47 | 0.39 |
| **L2P-PGP** | **97.53** | **0.17** | **98.58** | **0.28** |

# K  CONTINUAL LEARNING RESULTS ON MULTI-MODEL BACKBONE, COMPARISON BETWEEN CLIP-PGP WITH CLIP

We conduct our experiments on 10-Split-CIFAR100 dataset under class incremental setting and task incremental setting respectively, as shown in Table 8. Results show that, our method has improved the performance a lot for both the above settings, proving that our method is also useful in the vision-language models, which further enlarges the scope of our method.

Table 8: Comparison to *CLIP* model **with/without** gradient projection method on 10-Split-CIFAR100 with class/task incremental settings.

| Settings | Class Incremental | | Task Incremental | |
|---|---|---|---|---|
| Models | Accuracy | Forgetting | Accuracy | Forgetting |
| CLIP | 73.76 | 5.60 | 92.69 | 2.34 |
| **CLIP-PGP(Ours)** | **79.47(+5.71)** | **4.23(-1.37)** | **93.00(+0.31)** | **1.58(-0.76)** |

# L  CLASS INCREMENTAL LEARNING RESULTS ON DIFFERENT BACKBONES, COMPARISON BETWEEN OURS WITH BASELINES

To show the efficacy of proposed method on different pre-trained backbones, we evaluate our method by extending two distinct pre-trained models, namely ViT-DINO and ViT-SAM (Caron et al., 2021; Chen et al., 2021). The results are shown in the Table 9. Additionally, we tested our method on 10-Split-CIFAR100 and 5-Split-CUB200 dataset based on three pre-trained ViTs: ImageNet-21K, DINO, and SAM, further validating the effectiveness of our method on non-ImageNet datasets (Wah et al., 2011; Krizhevsky et al., 2009).

Table 9: Comparison to distinct pre-trained backbones between baselines and ours. **Red** parts show significant improvements ($>$**1**).

| Method | Pretrained-Dataset | 10-Split-CIFAR100 | | 5-Split-CUB200 | |
|---|---|---|---|---|---|
| | | ACC($\uparrow$) | Forgetting($\downarrow$) | ACC($\uparrow$) | Forgetting($\downarrow$) |
| L2P | ImageNet-21K | 83.77 | 6.63 | 74.88 | 5.39 |
| **L2P-PGP** | **ImageNet-21K** | **84.34(+0.57)** | **5.59(-1.04)** | **75.15(+0.27)** | **4.51(-0.88)** |
| DualPrompt | ImageNet-21K | 86.50 | 5.77 | 82.02 | 4.23 |
| **DualPrompt-PGP** | **ImageNet-21K** | **86.92(+0.42)** | **5.35(-0.42)** | **82.46(+0.44)** | **3.76(-0.47)** |
| L2P | SAM | 83.93 | 6.68 | 73.98 | 6.77 |
| **L2P-PGP** | **SAM** | **84.26(+0.33)** | **5.64(-1.04)** | **76.45(+2.47)** | **5.91(-0.86)** |
| DualPrompt | SAM | 86.11 | 6.08 | 82.02 | 4.73 |
| **DualPrompt-PGP** | **SAM** | **86.92(+0.81)** | **5.04(-1.04)** | **82.28(+0.26)** | **4.65(-0.08)** |
| L2P | DINO | 67.35 | 9.69 | 44.10 | 9.77 |
| **L2P-PGP** | **DINO** | **70.60(+3.25)** | **4.73(-4.96)** | **44.80(+0.70)** | **6.06(-3.71)** |
| DualPrompt | DINO | 64.18 | 23.87 | 50.88 | 10.10 |
| **DualPrompt-PGP** | **DINO** | **73.33(+9.15)** | **10.27(-13.60)** | **51.03(+0.15)** | **9.06(-1.04)** |

# M  PGP WITH PROMPT NUMBER AND PROMPT WIDTH

In this section, for L2P-PGP model, we set distinct parameters in prompt numbers and prompt widths on 10-Split-CIFAR100 dataset, and further validate the efficiency of prompt gradient projection method. Results are shown in Table 10. In our setting, we set a single prompt mode, that all tasks share a single prompt for training. We think, in this way, we can deeply uncover the potential of our method and avoid interference caused by choosing prompts. Results show that, models with prompt gradient projection, all have higher accuracy and lower forgetting than those without, which proves that our method could be effective in distinct prompt numbers and widths, even with a hard single prompt setting.

Table 10: Comparison L2P with L2P-PGP on 10-Split-CIFAR100 dataset. Width and number mean prompt width and prompt number respectively.

| | L2P | | L2P-PGP | | | L2P | | L2P-PGP | |
|---|---|---|---|---|---|---|---|---|---|
| Width | ACC(↑) | FOR(↓) | ACC(↑) | FOR(↓) | Number | ACC(↑) | FOR(↓) | ACC(↑) | FOR(↓) |
| 5 | 82.64 | 6.73 | 82.77 | 6.58 | 1 | 82.64 | 6.73 | 82.77 | 6.58 |
| 10 | 82.09 | 7.07 | 82.16 | 6.74 | 3 | 84.17 | 5.92 | 84.19 | 5.60 |
| 15 | 83.09 | 6.38 | 84.21 | 5.62 | 5 | 83.23 | 6.66 | 83.82 | 6.62 |
| 20 | 83.42 | 6.38 | 83.87 | 5.89 | 7 | 83.87 | 7.13 | 84.44 | 6.58 |
| 25 | 83.69 | 6.49 | 83.85 | 6.39 | 9 | 84.11 | 6.60 | 84.15 | 6.52 |
| 30 | 83.87 | 6.46 | 84.39 | 6.44 | | | | | |

# N    PGP WITH PREFIX WIDTH AND PREFIX PREPENDED LAYER

In this section, for DualPrompt-PGP model, we discuss whether prompt gradient projection could be efficient in different prefix widths and prepended layers. As the setting in Appendix M, we choose a single prefix mode based on the same reason. We conduct experiments on 10-Split-CIFAR100 and 10-Split-TinyImageNet. Final results are shown in Table 11 and Table 12. We also show some cases with curves of accuracy and forgetting metrics changing in all tasks, as in Figure 6 and Figure 7.

Table 11: Comparison DualPrompt with DualPrompt-PGP on 10-Split-CIFAR100 dataset. Width and layer mean prefix width and prefix prepended layer index respectively.

| | DualPrompt | | DualPrompt-PGP | | | DualPrompt | | DualPrompt-PGP | |
|---|---|---|---|---|---|---|---|---|---|
| Width | ACC(↑) | FOR(↓) | ACC(↑) | FOR(↓) | Layer | ACC(↑) | FOR(↓) | ACC(↑) | FOR(↓) |
| 5 | 81.08 | 7.64 | 81.49 | 7.08 | 0 | 81.08 | 7.64 | 81.49 | 7.08 |
| 6 | 81.32 | 7.12 | 81.70 | 6.89 | 0,1 | 82.22 | 5.78 | 82.75 | 5.67 |
| 7 | 81.67 | 7.51 | 81.95 | 6.77 | 0,1,2 | 83.85 | 5.62 | 84.69 | 4.38 |
| 8 | 81.67 | 7.48 | 81.92 | 7.06 | 0,1,2,3 | 84.55 | 5.03 | 84.58 | 4.84 |
| 9 | 81.74 | 6.49 | 81.88 | 6.21 | 0,1,2,3,4 | 84.59 | 5.60 | 84.74 | 5.04 |
| 10 | 81.58 | 6.93 | 81.63 | 6.78 | | | | | |

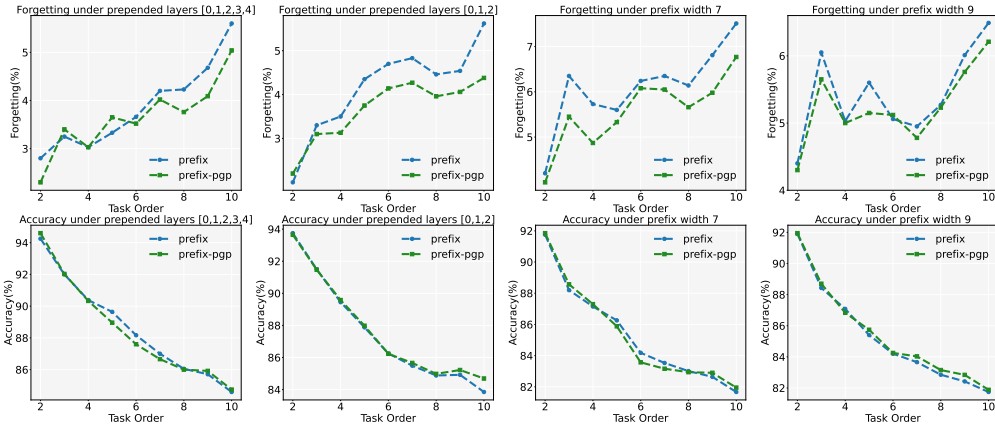

Figure 6: Changing curves of accuracy and forgetting metrics with different prepended layers and prefix widths on 10-Split-CIFAR100 dataset.

Results are similar to the discussion in Appendix M. Whether on 10-Split-CIFAR100 or 10-Split-TinyImageNet, models with prompt gradient projection always have better accuracy and lower forgetting than those without. We think it proves that our method can be effective in distinct prefix widths and prepended layers. Notice that we name the baseline as "prefix" and our method as "prefix-pgp".

Table 12: Comparison DualPrompt with DualPrompt-PGP in different settings on 10-Split-TinyImageNet dataset. Width and layer mean prefix width and prefix prepended layer index respectively.

| | DualPrompt | | DualPrompt-PGP | | | DualPrompt | | DualPrompt-PGP | |
|---|---|---|---|---|---|---|---|---|---|
| Width | ACC(↑) | FOR(↓) | ACC(↑) | FOR(↓) | Layer | ACC(↑) | FOR(↓) | ACC(↑) | FOR(↓) |
| 5 | 81.58 | 4.63 | 81.79 | 4.51 | 0 | 81.58 | 4.63 | 81.79 | 4.51 |
| 6 | 81.39 | 4.66 | 81.67 | 4.50 | 0,1 | 82.98 | 4.29 | 83.33 | 3.98 |
| 7 | 81.60 | 4.93 | 81.78 | 4.43 | 0,1,2 | 83.66 | 4.11 | 83.76 | 3.96 |
| 8 | 81.36 | 4.63 | 81.65 | 4.44 | 0,1,2,3 | 83.64 | 4.62 | 84.51 | 3.72 |
| 9 | 81.55 | 4.80 | 81.93 | 4.70 | 0,1,2,3,4 | 83.61 | 4.68 | 83.95 | 4.23 |
| 10 | 82.20 | 4.34 | 82.22 | 3.96 | | | | | |

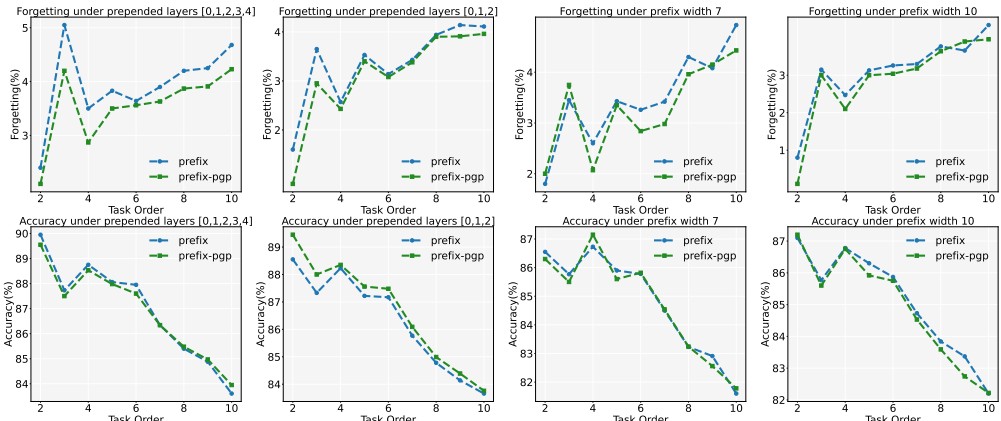

Figure 7: Changing curves of accuracy and forgetting metrics with different prepended layers and prefix widths on 10-Split-TinyImageNet dataset.

## O    T-SNE VISUALIZATION

To better visualize the improvement of our method, we choose L2P and L2P-PGP models. Training on 10-Split-CIFAR100 dataset, we show the T-SNE results of samples from task 1 across models in various tasks. We pick up logits processed by classifier to report.

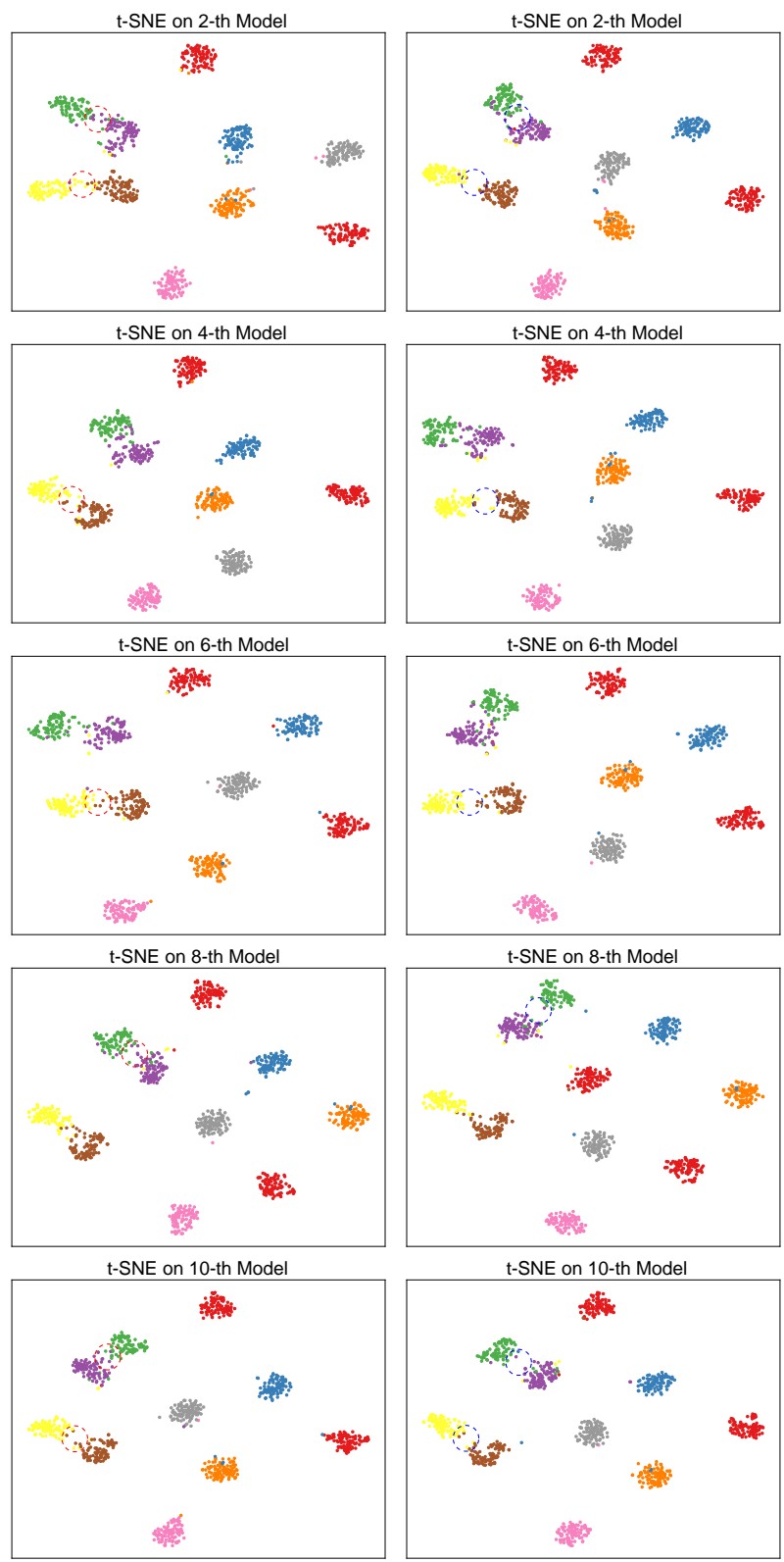

Figure 8: T-SNE results of L2P and L2P-PGP on 10-Split-CIFAR100 dataset. The left column represents L2P, and the right column represents L2P-PGP. The red circle means the drawback existing in L2P, and the blue circle shows the improvement of our method.

# P   ALGORITHM

---

**Algorithm 1:** Prompt Gradient Projection For L2P (Training phase)

---

**Input:** Pre-trained ViT model $f_\theta$, embedding layer $\phi_\theta$, classifier head $f_c$, number of tasks $T$, training set $\{\{X_i^t, y_i^t\}_{i=1}^{n_t}\}_{t=1}^T$, sampling set $\{\{X_{si}^t, y_{si}^t\}_{i=1}^{n_{st}}\}_{t=1}^T$, prompt pool $\{p_j\}_{j=1}^M$, projection matrix $V_{t,0}$, number of training epochs $E$, learning rate $\eta$, loss function $\mathcal{L}_x$

**Output:** prompt pool $\{p_j\}_{j=1}^M$, classifier head $f_c$

**initialize:** $f_c, \{p_j\}_{j=1}^M$.

**for** $t = 1, ..., T$ **do**

    **for** $e = 1, ..., E$ **do**

        Draw a mini-batch $B = \{(X_i^t, y_i^t)\}_{i=1}^{n_t}$.

        **for** $(X, y)$ *in* $B$ **do**

            Embed $X$ into sequence $x_t$ by $x_t = \phi_\theta(X)$.

            Select prompt $p_x$ from $\{p_j\}_{j=1}^M$.

            Prepend $x_t$ with $p_x$ by $x_p = [p_x; x_t]$.

            Obtain prediction by $\hat{y} = f_c(f_\theta(x_p))$.

        **end**

        Calculate per batch loss $\mathcal{L}_B$ by accumulating $\mathcal{L}_x(y, \hat{y})$.

        # Gradient projection

        **if** $t = 1$ **then**

            Update $p$ by $p \leftarrow p - \eta \nabla_p \mathcal{L}_B$.

        **else**

            Update $p$ by $p \leftarrow p - \eta \nabla_p \mathcal{L}_B V_{t,0} V_{t,0}^T$.

        **end**

    **end**

    # Gradient projection matrix update

    Initialize the sets of sampled embedding sequences and prompts: $X_t = \{\}, P_t = \{\}$.

    **for** $(X_{si}^t, y_{si}^t)$ *in* $\{(X_{si}^t, y_{si}^t)\}_{i=1}^{n_{st}}$ **do**

        Sample set of embedding sequences $X_t$ by concatenation of $X_t$ and $\phi_\theta(X_{si}^t)$.

    **end**

    **for** $p$ *in* $\{p_j\}_{j=1}^M$ *and* $p \in p_x$ **do**

        Sample set of prompts $P_t$ by concatenation of $P_t$ and $p$.

    **end**

    Update $V_{t,0}$ by $X_t$ and $P_t$ according to Appendix C.

**end**

---

**Algorithm 2:** Prompt Gradient Projection For L2P (Testing phase)

---

**Input:** Pre-trained ViT model $f_\theta$, embedding layer $\phi_\theta$, classifier head $f_c$, number of tasks $T$, test set $\{\{X_i^t\}_{i=1}^{n_t}\}_{t=1}^T$, prompt pool $\{p_j\}_{j=1}^M$

**Output:** prediction $\hat{y}$

**for** $t = 1, ..., T$ **do**

    **for** $X_i^t$ *in* $\{X_i^t\}_{i=1}^{n_t}$ **do**

        Embed $X_i^t$ into sequence $x_t$ by $x_t = \phi_\theta(X_i^t)$.

        Select prompt $p_x$ from $\{p_j\}_{j=1}^M$.

        Prepend $x_t$ with $p_x$ by $x_p = [p_x; x_t]$.

        Obtain prediction by $\hat{y} = f_c(f_\theta(x_p))$.

    **end**

**end**

---

