# OpenReview forum: "Prompt Gradient Projection for Continual Learning"
_ICLR.cc/2024/Conference — ICLR 2024 spotlight_

### Official Review · Reviewer_LEcv · 2023-10-30

**Soundness:** 3 good
**Presentation:** 4 excellent
**Contribution:** 2 fair
**Rating:** 6
**Confidence:** 3

**Summary:**

The paper proposes three steps to derive the prompt gradient projection (PGP) approach for continual learning of pre-trained models with prompt tuning. First, using the self-attention mechanism in ViT, the work deduces the gradient restriction conditions for prompt learning based on the feature and the prompt vectors. Second, to simply the solving of these conditions, the work uses a sum space of the feature and the prompt vectors and conducts Singular Value Decomposition (SVD) on this space. Finally, PGP exploits balancing plasticity and stability by rearranging the singular values split on the basis of a threshold.

**Strengths:**

Given the rising popularity of using pre-trained models for downstream tasks, this is a timely topic to study how such models can be adapted for effective CL.

- Clear explanations and equations built up from scratch.
- Reasonable experimental settings.

**Weaknesses:**

- Given that the L2P and DualPrompt backbones have been pre-trained on ImageNet, the limited performance gain on the 10-Split-TinyImageNet is a bit concerning (Table 3). Also, the gains on other dataset settings follow a similar trend. Alternatively, the work could have used a backbone trained on a different dataset altogether [1] and then evaluated its performance on the said settings. A third option would be to use other CL datasets like CUB-200, ObjectNet, etc. that have limited domain and style overlap with the ImageNet dataset.
- The work claims that one of the major advantages of the proposed PGP method is its training time and memory cost. However, Table 2 does not report the training time of the baseline (L2P-R). As such, it is unclear what conclusions can be derived about the training time.
- What concerns me is the scope of the proposed PGP method. While the authors use the classic L2P and DualPrompt as their baselines, it would have been more interesting to see how the proposed method complements the continual learning of more powerful pre-trained models such as vision-language models [2]. Also, the baselines considered in this paper are rather weak. Many latest methods are missing. Such as LoRA, K-Adapter, and other parameter-expansion methods.

Minor comments:

- Definition of V_t missing - the paper introduces it as a two part column vector (before eq. 11) without explaining what does it contain. The same for U_t (prior to eq. 10).

References:

[1] Zhou, Da-Wei et al. “Learning without Forgetting for Vision-Language Models.” ArXiv abs/2305.19270 (2023): n. pag.

[2] Thengane, Vishal G. et al. “CLIP model is an Efficient Continual Learner.” ArXiv abs/2210.03114 (2022): n. pag.

**Questions:**

Please see the weaknesses. Overall, it is a good incremental contribution to the field of prompt-tuning for continual learning. However, I am worried about the limited efficacy of the proposed method as well as the scope of the reported experimental baselines. Therefore, I cannot recommend an acceptance.

---

> ### Author Response · Authors · 2023-11-20
> **Response to Reviewer LEcv For Weakness 1**
>
> Dear reviewer LEcv, thanks for your valuable suggestions. Here are our responses:
>
> For **Weakness 1** about *''use a backbone trained on a different dataset and use other CL datasets''*:
>
> As your suggestion, we re-evaluated the effectiveness of our method by **extending two distinct pre-trained models**, namely **ViT-DINO** and **ViT-SAM**. The results are shown in the Table below. Additionally, we tested our method on the **CUB-200** dataset based on three ViT pre-trained backbones: ImageNet-1k, DINO, and SAM, further validating the effectiveness of our method on non-ImageNet datasets.
>
> **Table 1**: Comparison to distinct pretrained-dataset backbones with/without prompt gradient projection.
> |                          |                    | 10-Split-CIFAR100                                    | 10-Split-CIFAR100                                     | 5-Split-CUB200                                       | 5-Split-CUB200                                      |
> | ------------------------ | ------------------ | ---------------------------------------------------- | ----------------------------------------------------- | ---------------------------------------------------- | --------------------------------------------------- |
> | Method                   | Pretrained-Dataset | Accuracy                                             | Forgetting                                            | Accuracy                                             | Forgetting                                          |
> | L2P                      | DINO[1]            | 67.35                                                | 9.69                                                  | 44.10                                                | 9.77                                                |
> | **L2P-PGP(Ours)**        | **DINO**           | **70.60*****(+3.25)*** | **4.73*****(-4.96)***   | **44.80*****(+0.70)***                     | **6.06*****(-3.71)*** |
> | DualPrompt               | DINO               | 64.18                                                | 23.87                                                 | 50.88                                                | 10.10                                               |
> | **DualPrompt-PGP(Ours)** | **DINO**           | **73.33*****(+9.15)*** | **10.27*****(-13.60)*** | **51.03*****(+0.15)***                     | **9.06*****(-1.04)*** |
> | L2P                      | SAM[2]             | 83.93                                                | 6.68                                                  | 73.98                                                | 6.77                                                |
> | **L2P-PGP(Ours**         | **SAM**            | **84.26*****(+0.33)***                     | **5.64*****(-1.04)***   | **76.45*****(+2.55)*** | **5.91*****(-0.86)***                     |
> | DualPrompt               | SAM                | 86.11                                                | 6.08                                                  | 82.02                                                | 4.73                                                |
> | **DualPrompt-PGP(Ours)** | **SAM**            | **86.92*****(+0.81)***                     | **5.04*****(-1.04)***   | **82.28*****(+0.26)***                     | **4.65*****(-0.08)***                     |
> | L2P                      | ImageNet-1K        | 83.77                                                | 6.63                                                  | 74.88                                                | 5.39                                                |
> | **L2P-PGP(Ours)**        | **ImageNet-1K**    | **84.34*****(+0.57)***                               | **5.59*****(-1.04)***   | **75.15*****(+0.27)***                     | **4.51*****(-0.88)***                     |
> | DualPrompt               | ImageNet-1K        | 86.50                                                | 5.77                                                  | 82.02                                                | 4.23                                                |
> | **DualPrompt-PGP(Ours)** | **ImageNet-1K**    | **86.92*****(+0.42)***                     | **5.35*****(-0.42)***                       | **82.46*****(+0.44)***                     | **3.76*****(-0.47)***                     |
>
> [1] DINO pre-trained models (https://arxiv.org/abs/2104.14294)
>
> [2] SAM pre-trained models (https://arxiv.org/abs/2106.01548)

---

> ### Author Response · Authors · 2023-11-20
> **Response to Reviewer LEcv For Weakness 2**
>
> For **Weakness 2** about *''the training time consumption''*:
>
> To answer your question, we reconducted experiments about L2P with data rehearsal on the 10-Split-CIFAR100 dataset. we randomly sampled the exemplars and achieved a result akin to the report results in the paper. Then, we measured the time consumption of our reproduced L2P-R and compared it with our method **as shown in Table 2**. The time consumption of *our method* is **0.756 h** and the time consumption of *L2P-R* is **0.787 h**. The reason why our method is faster than L2P-R, we hold the view is that L2P-R needs to train the extra rehearsal data in every epoch, which spends a lot of time. The more rehearsal data is saved, the more training time is consumed. However, our method is rehearsal-free.
>
> **Table 2**: Time Complexity Comparison, Accuracy, Forgetting between our approach and L2P, L2P-R. L2P-R indicates L2P with data rehearsal.
> | Method            | Exemplar | Time        | Accuracy  | Forgetting |
> | ----------------- | -------- | ----------- | --------- | ---------- |
> | L2P               | 0        | 0.74 h      | 83.77     | 6.63       |
> | L2P-R             | 1000     | 0.787 h     | 84.21     | 7.72       |
> | **L2P-PGP(Ours)** | **0**    | **0.756 h** | **84.34** | **5.59**   |
>
> To sum up, our method not only **exhibits superior performance** and **requires less storage memory** but also **consumes less training time**.

---

> ### Author Response · Authors · 2023-11-20
> **Response to Reviewer LEcv For Weakness 3**
>
> For **Weakness 3** about *''the scope of the proposed PGP method''*:
>
> To further validate our method on the vision-language model, we introduce it to the **CLIP** model. The experimental setting is that, in the vision side, we only set a single trainable image prompt shared by all tasks. As for the text side, we follow the operation as [3], we set trainable text prompt for each class, which is only trained at the corresponding task. We conduct our experiments on two datasets: *10-Split-CIFAR100* and *10-Split-TinyImageNet*, **as shown in Table 3**. Besides that, we also verify the effectiveness of our method under the task incremental setting, **as shown in Table 4**. Results show that our method has improved the performance a lot for all the above settings, which we think proves that our method is also **useful in the vision-language models and further enlarges the scope of our method**.
>
> We have noticed that, compared with your mentioned method, our method does not exhibit comparable accuracy performance, despite conducting experiments on entirely different datasets with [4]. The reasons are as follows: **Firstly**, in comparison to its **complex method and network structure**, we employ a **very simple setting**, utilizing only a single image prompt, which restricts the performance of our method. **Secondly**, it **employs the data rehearsal method** to further enhance performance. However, our method is **rehearsal-free**.
>
> In our design, we want to exclude all of the external influential factors and only verify the PGP method. Thus, we choose the simplest experiment setting of a single image prompt. Experiments show that our method works well.
>
> **Table 3**: Comparison to *CLIP* model without/with gradient projection method under **class incremental setting** (**without** task identifier in the test phase).
> |                    | 10-Split-CIFAR100 | 10-Split-CIFAR100 | 10-Split-TinyImageNet | 10-Split-TinyImageNet |
> | ------------------ | ----------------- | ----------------- | --------------------- | --------------------- |
> | Models             | Accuracy          | Forgetting        | Accuracy              | Forgetting            |
> | CLIP               | 58.95             | 6.29              | 56.28                 | 7.19                  |
> | **CLIP-PGP(ours)** | **63.72*****(+4.77)*** | **5.09*****(-1.20)***  | **60.34*****(+4.06)***     | **6.21*****(-0.98)***      |
>
> **Table 4**: Comparison to *CLIP* model without/with gradient projection method under **task incremental setting** (**with** task identifier in the test phase).
> | Models             | Accuracy               | Forgetting            |
> | ------------------ | ---------------------- | --------------------- |
> | CLIP               | 92.69                  | 2.34                  |
> | **CLIP-PGP(ours)** | **93.00*****(+0.31)*** | **1.58*****(-0.76)*** |
>
> As for the latest methods, such as LoRA, K-Adapter, and other parameter-expansion methods. Since they mainly focus on fine-tuning a large model, whose goal is to adapt the model to the down-stream tasks, we are afraid this is beyond the scope of this paper, which aims to learn new concepts while not forgetting the prompts. Also, the combination of gradient projection and fine-tuning methods would be interesting. We would like to study this in the future.
>
> [3] Zhou, Kaiyang, et al. "Learning to prompt for vision-language models." International Journal of Computer Vision 130.9 (2022): 2337-2348.
>
> [4] Zhou, Da-Wei et al. “Learning without Forgetting for Vision-Language Models.” ArXiv abs/2305.19270 (2023): n. pag.

---

> ### Author Response · Authors · 2023-11-20
> **Response to Reviewer LEcv For Minor Comments**
>
> For **minor comments** about *''Definition of V_t and U_t missing''*:
>
> Thanks for your valuable suggestions. We have **added the missing definition of** ***V_t*** **and** ***U_t***.

---

> ### Author Response · Authors · 2023-11-20
> **Response to Reviewer LEcv For Questions**
>
> To answer **your concerns about our method of the limited efficacy and the scope of the reported experimental baselines**, we conclude our method as follows:
>
> **(1)** **We are the first one to propose the preventing forgetting method from the view of prompt gradient**. As continual learning is not a fine-tuning task, the data is coming as a sequence of tasks, and data from previous tasks would not be seen in future tasks, which brings a heavy decline in performance of models. Thus, how to overcome catastrophic forgetting is the prime aim of continual learning. To the best of our known, **we are the first one to deduct the orthogonal condition to prevent forgetting for prompt-based methods and implement the gradient projection on prompt**. We believe that our method owns a huge potential, which is still uncovered. In future works, we will **further explore the potential of our method to achieve better results, and also release the code**.
>
> **(2)** We conduct our method on multiple and various models, *e,g, ViT, CLIP*, distinct prompt-tuning paradigms, *e,g, prompt-tuning, prefix-tuning*, different pre-trained backbones, *e.g. ImageNet-1K, SAM, and DINO*, under three continual learning settings: *class incremental learning, task incremental learning, and online class incremental learning*. **All experiments show that our method can effectively reduce forgetting and improve the average accuracy**. Besides that, **our method is a plug-in, and can serve in different prompt-based continual learning methods**. Especially nowadays, as the tendency of large models become hotter and hotter, we think **our method can have a promising application future**.
>
> **(3)** We prove that **our method has little time and memory consumption** through experiments. Besides that, our method does out introduce other models or save any exemplars. Based on these facts, we believe that our method can have advantages in training speed and memory, which **could be beneficial when transferring our method to large vision foundation models (VFMs)**.
>
> **(4)** Finally, **we reveal the mechanism of trade-off between the plasticity and stability in the perspective of prompt gradient, and well balance the plasticity and stability of model by adjusting the threshold *\epsilon***, which brings a new insight as the above trade-off dilemma is a core problem in continual learning.

---

> > ### Comment · Reviewer_LEcv · 2023-11-21
> >
> > Thank you for the clarification. In light of the additional experimental results and explanations of the work's scope, I have decided to increase my rating to 6 (from 3).

---

> > > ### Author Response · Authors · 2023-11-21
> > > **Thanks for Reviewer LEcv**
> > >
> > > Thank you for your hard work in reviewing our paper. We sincerely appreciate your valuable comments and suggestions, such as applying our method to distinct pre-trained backbones and the vision-language model, which enhance the quality of our paper.

---

### Official Review · Reviewer_epe6 · 2023-10-30

**Soundness:** 3 good
**Presentation:** 2 fair
**Contribution:** 3 good
**Rating:** 8
**Confidence:** 4

**Summary:**

The paper presents Prompt Gradient Projection (PGP), a novel approach that combines Prompt Tuning with Gradient Projection to address the challenge of forgetting in continual learning. Prompt Tuning reduces forgetting in class-incremental learning by selecting and updating relevant prompts based on input samples, while Gradient Projection prevents forgetting in task-incremental learning by ensuring gradient updates in orthogonal directions to old features. PGP integrates these techniques, releasing the need for task identifiers in Gradient Projection and providing theoretical guarantees against forgetting. By deducing the orthogonal condition for anti-forgetting in prompt gradients and using Singular Value Decomposition for efficient computation, PGP significantly reduces forgetting and improves accuracy in various learning settings, achieving state-of-the-art results on benchmark datasets.

**Strengths:**

The paper introduces a novel approach, Prompt Gradient Projection (PGP), which combines Prompt Tuning and Gradient Projection to tackle the issue of forgetting in continual learning. The problem statement is well-defined, and related work is thoroughly reviewed, showcasing a strong understanding of the research landscape. The paper's strength lies in its extensive and insightful experimental studies, including ablation studies that highlight the efficiency of PGP and the influence of various factors. Furthermore, the inclusion of proofs, extensive experiments, and detailed algorithms in the appendix enhances the paper's credibility and reproducibility.

**Weaknesses:**

The clarity and readability of the current paper are areas of concern. The paper's overall structure and writing could be improved to enhance its accessibility. Several specific issues were identified:

Readability and Clarity: The paper's overall readability and clarity could be enhanced. Specific instances of unclear language were noted, such as in the abstract and the last paragraph of page 3. Further revisions are needed to make these sections more lucid.

Figure 2: Figure 2 was found to be unclear and challenging to follow, especially for readers seeking to grasp the method's essence. Consider revising or providing additional explanatory details to improve the comprehensibility of this figure.

Proof Presentation: While the proof on page 5 is well-explained up to equation 12, the subsequent section, where multiple elements are combined, is challenging to follow. Expanding on the last two paragraphs on page 5 could enhance clarity and understanding.

Terminology and Consistency: In page 6, there is mention of dividing $V_t$ into $V_{t,1}$ and $V_{t,2}$" while also referencing $V_{t,0}$". Clarification is needed to ensure consistency and prevent potential confusion, possibly by referring back to the explanation on page 5.

Notation Clarification: On page 6, there is a reference to "$V_{t,}$" which requires clarification to make it more understandable to the reader. Providing a clear definition or context for this notation would be beneficial.

Overall, addressing these issues will significantly improve the paper's accessibility and enhance its overall quality, making it more suitable for publication.

**Questions:**

The paper raises several important points that require clarification and further exploration:

Reference Request: In the paragraph preceding equation 1, the statement "If the update direction is orthogonal to the old features, it follows that $\Delta W_t x_{t,i}=0$" lacks a specific reference. It would be beneficial to provide a reference or additional context to support this claim and enhance the paper's credibility.

Time Complexity: The paper would benefit from a more detailed discussion of the method's time complexity, along with a comparative analysis against relevant baseline methods. Understanding the computational efficiency of the proposed approach in relation to other methods is crucial for assessing its practical utility.

Performance vs. Time Complexity Trade-off: The paper mentions a maximum improvement of around 1% in reducing catastrophic forgetting in the reported experimental results. It is essential to provide a more in-depth discussion of the trade-off between the promising performance of the proposed algorithm and its associated time complexity. Explaining why this algorithm should be prioritized over other existing algorithms, considering the modest improvement, would provide valuable insights.

Figure 3 Clarification: Figure 3 requires further elaboration. While it is presented that both algorithms exhibit similar patterns, with Dual-Prompt appearing to be shifted up(accuracy) or down (forgetting) by a certain scale, a more detailed explanation of this observation is necessary. Clarifying the significance of these patterns and the implications for the proposed method's effectiveness is essential.

Addressing these concerns will contribute to a clearer and more comprehensive understanding of the paper's content and its contributions to the field.

================================
I appreciate the authors for offering a thorough rebuttal!
I find this paper to be intriguing and innovative, leading me to raise my score to 8.

---

> ### Author Response · Authors · 2023-11-20
> **Response to Reviewer epe6 For Weakness 1**
>
> Dear reviewer epe6, thanks for your valuable suggestions. Here are our responses:
>
> For **Weakness 1** about *''Readability and Clarity''*:
>
> We have rewritten the abstract and the last paragraph of page 3, and further revised several sections of the paper where language ambiguity exists. We thank your suggestion which makes our paper clearer and more readable than before. The following are two examples of revised version:
>
> *For Abstract*
>
> Before revised: Prompt-tuning has demonstrated impressive performance in continual learning by querying relevant prompts for novel classes training.
>
> **After revised**: Prompt-tuning has demonstrated impressive performance in continual learning by querying relevant prompts for each input instance, which can avoid the introduction of task identifier.
>
> Before revised: In PGP, we deduce the orthogonal condition for prompt gradient via the self-attention mechanism in vision-transformer.
>
> **After revised**: In PGP, we deduce that reaching the orthogonal condition for prompt gradient can effectively prevent forgetting via the self-attention mechanism in vision-transformer.
>
> *For the last paragraph of page 3*
>
> Before revised: one limitation of gradient projection methods is which is only applicable for task incremental learning but will fail in the class-incremental inference, as the projected gradient is a strict constraint for novel classes training.
>
> **After revised**: one limitation of gradient projection methods fail in the class-incremental inference is that the projected gradient needs task identifier to find relevant network parameters.

---

> ### Author Response · Authors · 2023-11-20
> **Response to Reviewer epe6 For Weakness 2**
>
> For **Weakness 2** about *''Figure 2''*:
>
> We have **enhanced the clarity** of *Figure 2* by providing a more detailed explanation in the caption, elucidating the content and essence of our method, and **further improving the comprehensibility** of the image.

---

> ### Author Response · Authors · 2023-11-20
> **Response to Reviewer epe6 For Weakness 3**
>
> For **Weakness 3** about *''Proof Presentation''*:
>
> We have **expanded** the content in the *last two paragraphs* of *page 5*, providing a more compact connection with the preceding and following sections. This adjustment is intended to enhance the clarity and comprehensibility of the paper.

---

> ### Author Response · Authors · 2023-11-20
> **Response to Reviewer epe6 For Weakness 4 and Weakness 5**
>
> For **Weakness 4** about *''Terminology and Consistency''*:
>
> In the revised version, we have **replaced** *V_t, V_1, and V_2* with **new symbolic representations** and **added detailed explanatory text** to ensure consistency with the preceding context and eliminate any potential confusion.
>
> For **Weakness 5** about *''Notation Clarification''*:
>
> Thank you for your question. It has been **addressed along with the previous one**.

---

> ### Author Response · Authors · 2023-11-20
> **Response to Reviewer epe6 For Question 1**
>
> For **Question 1** about *''Reference Request''*:
>
> We have inserted the following two pertinent references into the statement location as advised.
>
> [1] Saha, Gobinda, Isha Garg, and Kaushik Roy. "Gradient Projection Memory for Continual Learning." International Conference on Learning Representations. 2020.
>
> [2] LiLin, Sen, et al. "TRGP: Trust Region Gradient Projection for Continual Learning." International Conference on Learning Representations. 2021.

---

> ### Author Response · Authors · 2023-11-20
> **Response to Reviewer epe6 For Question 2**
>
> For **Question 2** about *''Time Complexity''*:
>
> Thanks for pointing out our omission.
>
> Under the same hardware platform, experimental setting, and backbone, we compared our method with the original L2P and L2P with data rehearsal on the *10-Split-CIFAR100* dataset. As L2P does not mention detailed information about how to make data rehearsal, we adopted a simple data rehearsal process which randomly sampled a set of exemplars and achieved a result similar to the reported ones.
>
> The results indicate that, in comparison with the *original L2P*, our approach reduces forgetting by 1.04%, improves accuracy by 0.57%, and *extends* training time by *0.016* hours. In comparison with **L2P with data rehearsal**, our method not only reduces forgetting by 2.13%, improves accuracy by 0.13%, but also **shortens** training time by **0.031** hours.
>
> **Table 1**: Time Complexity Comparison between our approach and L2P, L2P-R. L2P-R indicates L2P with data rehearsal.
> | Method            | Exemplar | Accuracy  | Forgetting | Time        |
> | ----------------- | -------- | --------- | ---------- | ----------- |
> | L2P               | 0        | 83.77     | 6.63       | 0.74 h      |
> | L2P-R             | 1000     | 84.21     | 7.72       | 0.787 h     |
> | **L2P-PGP(Ours)** | **0**    | **84.34** | **5.59**   | **0.756 h** |
>
> For further analysis, we use **“time.time()”** function to coarse read the time consumption in each sub-process.
>
> **For data rehearsal**, it concludes in three steps: **1. Collect exemplars (4\*10-5~26\*10-5 s)**, which can be ignored. **2. Store exemplars (Simplified in our reimplement method)**. **3. Exemplars join in the training process**. We mainly discuss the time consumption in the third step: In the batch size of 20, one iteration that only exemplars take part in costs a mean time of: 0.196 s. Here, we set epochs as 5 in the experiment. Thus, on the 10-Split-CIFAR100, training with exemplars needs extra (5+10+15+20+25+30+35+40+45) * 5 = 1125 iterations and costs 1125 * 0.196 = **220.50 s**. To sum up, theoretical analysis shows that data rehearsal would add an extra **220.50 s** in the training process.
>
> **For our method**, it also concludes three steps: **1. Collect samples (10-6\*33\*5\*10 s), which can be ignored**. **2. Calculate the projection matrix**, consumption time is (10.65 + 8.78 + 9.42 + 8.72 + 8.68 + 8.68 + 8.53 + 8.81 + 8.72 + 8.69 = 81 s). 3. **Gradient projection process**, consumption time is (2.79 + 3.22 + 2.95 + 2.73 + 3.11 + 3.28 + 2.72 + 3.01 + 2.78 + 2.98 = 29.57 s). To sum up, theoretical analysis shows that gradient projection would add an extra 81 + 29.57 = **110.57 s** in the training process.
>
> Notice that, we set 5 epochs in the training process. If we increase the epoch number, the consumption time of L2P-R would also increase due to more iterations of training exemplars. However, our method is unrelated to the epoch number and can be free of increasing epochs.

---

> ### Author Response · Authors · 2023-11-20
> **Response to Reviewer epe6 For Question 3**
>
> For **Question 3** about *''Performance vs. Time Complexity Trade-off''*:
>
> Thanks for your valuable suggestions.
>
> (1) Discussion of the promising performance of the proposed algorithm and its associated time complexity:
>
> **i)** For Time: In the answer to time complexity, we have shown detailed information about time consumption in our method. In fact, both the time consumption on calculation of projection matrix and gradient projection process is little, only adding **2%** extra training time on the original baseline.
>
> **ii)** For Performance: Our approach demonstrates a high generalization ability, could be combined with the almost prompt-based continual learning methods. Besides that, from the perspective of gradient projection, we further realize a well-done balance of the model's plasticity and stability. This balance is crucial for continual learning, causing a favorable balance often results in enhanced performance.
>
> (2) Explaining why this algorithm should be prioritized over other existing algorithms:
>
> To the best of our knowledge, we are the first to propose the gradient projection for reducing forgetting in the context of prompt-tuning. Note that the tendency of vision foundation model has recently caused significant changes in computer vision. Our approach first studies the gradient of prompt-learning and we believe related applications, like fine-tuning a VFM, could benefit from our paper. **We have additionally tested the large vision foundation model for continual learning, and the results turn out our approach is also effective in this scenario. Please kindly refer to Table 2**. Here, for the vision side, we set a single trainable image prompt shared by each task. For the text side, we follow the operation with [3], we set trainable text prompt for each class, only trained at the related task.
>
> **Table 2**: Comparison to *CLIP* model without/with gradient projection method on 10-Split-CIFAR100 with **class incremental setting** (**without** task identifier in the test phase).
> |                    | 10-Split-CIFAR100  | 10-Split-CIFAR100 | 10-Split-TinyImageNet | 10-Split-TinyImageNet |
> | ------------------ | ------------------ | ----------------- | --------------------- | --------------------- |
> | Models             | Accuracy           | Forgetting        | Accuracy              | Forgetting            |
> | CLIP               | 58.95              | 6.29              | 56.28                 | 7.19                  |
> | **CLIP-PGP(ours)** | **63.72*****(+4.77)*** | **5.09*****(-1.20)*** | **60.34*****(+4.06)***    | **6.21*****(-0.98)***     |
>
> Here, we also compare our method with other **two non-gradient and prompt-based methods**, one is data rehearsal, and the other is a paper titled "Introducing Language Guidance in Prompt-based Continual Learning", referred to as **LGCL**, presented at ICCV’23 [4]. LGCL is slightly lower than our method but with complicated designs, **as shown in Table 3**. Compared with baseline and IGCL, we have found we own the following three advantages.
>
> **1.** For memory, we only need to store a tiny projection matrix. While for data rehearsal, saving exemplars would spend far more memory space than us. For LGCL, although it is a rehearsal-free method, it has to introduce another network, a pre-trained text encoder: CLIP L/14 into the method, which also needs to occupy a significant amount of memory space.
>
> **2.** For time, extra time in our method, mainly costs on calculation of projection matrix and gradient projection process, which is shorter than the retraining process of exemplars in the data rehearsal method, explained in the answer of time complexity. For LGCL, due to the non-open source code, we cannot reproduce the experiments. However, in the paper, it adds an additional text encoding process and adopts extra loss functions to train the model. Therefore, compared to LGCL, our method also exhibits shorter time complexity.
>
> **3.** Our method could serve as a plug-in to reduce forgetting of the models in almost all of the prompt-based continual learning methods. With the hot tendency of the large models nowadays, our method could be transferred to the large Vision Foundation Models (VFMs) and have a promising application future in the fine-tuning paradigm.
>
> **Table 3**: Comparison to Accuracy, Forgetting of L2P-LGCL and L2P-PGP.
> |                   | 10-Split-CIFAR100 | 10-Split-CIFAR100 |
> | ----------------- | ----------------- | ----------------- |
> | Method            | Accuracy          | Forgetting        |
> | L2P-LGCL          | 84.33±0.06        | 5.83±0.23         |
> | **L2P-PGP(Ours)** | **84.34±0.08**    | **5.59±0.05**     |
>
> [3] Zhou, Kaiyang, et al. "Learning to prompt for vision-language models." International Journal of Computer Vision 130.9 (2022): 2337-2348.
>
> [4] Khan, Muhammad Gul Zain Ali, et al. "Introducing Language Guidance in Prompt-based Continual Learning." Proceedings of the IEEE/CVF International Conference on Computer Vision. 2023.

---

> ### Author Response · Authors · 2023-11-20
> **Response to Reviewer epe6 For Question 4**
>
> For **Question 4** about *''Figure 3 Clarification''*:
>
> Our explanation for this issue is as follows: Firstly, as you can see, Figure 3 is set in the background of online class-incremental learning, strictly limiting each image appearing only once. That is to say, each task is only trained for one epoch. The limited number of epochs results in fewer gradient update processes and fewer gradient projection processes, leading to **insufficient gradient projection**. Here, we emphasize that it is precisely due to the insufficient gradient projection that the phenomenon of "Dual-Prompt appearing to be shifted up (accuracy) or down (forgetting) by a certain scale" occurs.
>
> Our rationale is that each projection serves as a correction to the direction of gradient, thereby reducing forgetting. The more projections, the more sufficient the correction, and the more stable and effective the mitigation of forgetting. Conversely, fewer projections lead to insufficient correction, causing the mitigation of forgetting to be highly unstable, oscillating randomly between good and bad states of suppressing forgetting. The term *"a certain scale"* represents this **manifestation of random oscillation**. To demonstrate our conclusion, we conducted experiments under two settings: **epoch=1** and **epoch=3** on the same *10-Split-TinyImageNet* dataset, **as shown in Table 5 below**:
>
> The **"Diff"** in the Table represents **the difference between FOR (forgetting) and ACC (accuracy) after projection and before projection**. It can be observed that, at *epoch=1*, **the differences in ACC and FOR across different tasks fluctuate more noticeably, aligning with the oscillating process described in the theoretical explanation, showing an oscillating pattern between good and bad states**. However, at *epoch=3*, **the differences in ACC and FOR increase steadily with the increasing number of tasks, and the variations become highly stable**.
>
> **Table 5**: Comparison to the differences in ACC(Diff_ACC), the differences in FOR(Diff_FOR) at different tasks under settings of distinct epoch.
> | Task    | Epoch=1  | Epoch=1  | Epoch=3  | Epoch=3  |
> | ------- | -------- | -------- | -------- | -------- |
> | Task=1  | Diff_ACC | Diff_FOR | Diff_ACC | Diff_FOR |
> | Task=2  | 0.45     | 0.90     | 1.40     | 0.3      |
> | Task=3  | 1.40     | 0.83     | 1.45     | 0.52     |
> | Task=4  | 1.00     | 1.70     | 1.52     | 0.66     |
> | Task=5  | 1.34     | 1.08     | 1.65     | 0.90     |
> | Task=6  | 1.06     | 0.90     | 1.65     | 0.88     |
> | Task=7  | 0.55     | 0.53     | 1.70     | 0.89     |
> | Task=8  | 0.72     | 0.70     | 1.73     | 0.87     |
> | Task=9  | 0.94     | 0.91     | 1.80     | 1.14     |
> | Task=10 | 0.78     | 0.64     | 1.94     | 1.26     |

---

> ### Author Response · Authors · 2023-11-21
> **Thanks for Reviewer epe6**
>
> Thank you for your diligent review of our paper. With your valuable comments and suggestions, especially for the Questions about paper revision, we have significantly improved the clarity, fluency, and details of our paper. We sincerely appreciate you and your advice.

---

### Official Review · Reviewer_Q1LH · 2023-11-01

**Soundness:** 3 good
**Presentation:** 3 good
**Contribution:** 3 good
**Rating:** 8
**Confidence:** 5

**Summary:**

The submission proposes to combine prompt-tuning based continual learning methods with gradient projection methods. The submission explains how to project the gradient of the learnable prompt tokens and prompt key such that there is no forgetting. The evaluation is performed on 3 datasets and various different continual learning setups.

**Strengths:**

- Combining prompt tuning with gradient projection and therefore gaining understanding of the learnable prompts' gradient space in context of CL is novel and important
- The gradient projection derivation is sound and clearly explained.
- Evaluation is done on multiple datasets and settings, and the proposed method shows superiority against competitors such as L2P and DualPrompt.

**Weaknesses:**

- The comparison of prompt-tuning based CL methods with other CL methods does not seem fair to me. Prompt-tuning based methods use pretrained backbones which will give an performance edge when comparing against non-prompt-tuning based CL methods.

=========== Post-rebuttal changes ========
- The authors have resolved my concern on this particular issue.

**Questions:**

- Figure 1 is a bit unintuitive as normally, models that fill the radar chart are considered to be more powerful but in this case for FOR metrics being close to the center is better and for ACC metrics being fuller is better. Changing the FOG figure so higher is better would make the Figure more readable in my opinion.

- From what I could understand, the gradients for the learnable prompt tokens and prompt keys in the prompt pool are projected. What about gradients for the classifier attached to the backbone? Are these modified as well?

---

> ### Author Response · Authors · 2023-11-20
> **Response to Reviewer Q1LH For Weakness 1**
>
> Dear reviewer Q1LH, thanks for your valuable suggestions. Here are our responses:
>
> For **Weakness 1** about *''The comparison of prompt-tuning based CL methods with other CL methods''*:
>
> **Thanks for the advice. The backbone is actually pre-trained on ImageNet, and therefore it is somewhat unfair to compare with non-prompt-tuning based CL methods.**
>
> **However**, the tendency of the pre-trained large models, or vision foundation model has recently caused significant changes in computer vision. This paper focuses on reducing the forgetting of prompt-tuning paradigm, which, we believe, would bring new insights for continual learning. **We have additionally tested a lot of other baselines, especially including large vision foundation model, and the results turn out our approach is also effective in this scenario. Please kindly refer to Table 1.** Here, the methods in Table 1 are from 1) Replacing the pre-trained backbone of ImageNet-1K in L2P and DualPrompt with SAM and DINO. 2) Introducing prompt gradient projection method to CLIP model. For the vision side, we set a single trainable image prompt shared by each task. For the text side, following the operation with [1], we set trainable text prompt for each class, which is only trained at the corresponding task.
>
> **For the unfair comparison, please kindly note that:** **1)**, in our baseline (L2P and DualPrompt  [2, 3]), they have already compared their methods (adopting pre-trained backbones) with non-prompt-tuning-based CL methods (e,g. LwF, EWC, BiC, DER). We just simply follow their existing practice. We would denote the backbone comparison in our revised paper. **2)**, Our goal is to analyze the gradient space for prompt-tuning-based CL methods and propose a gradient projection method that effectively reduces forgetting. Therefore, our focus is mainly on observing performance improvements compared to the baselines (L2P and DualPrompt), non-prompt-tuning based CL methods are used as references.
>
> **Table 1:** Comparison to distinct baselines with/without prompt gradient projection method on 10-Split-CIFAR100.
>
> | Models               | Pre-trained backbone | Accuracy                                | Forgetting                               |
> | -------------------- | -------------------- | --------------------------------------- | ---------------------------------------- |
> | L2P                  | DINO[4]              | 67.35                                   | 9.69                                     |
> | **L2P-PGP(ours)**    | **DINO**             | **70.60*****(+3.25)*** | **4.73*****(-4.96)***   |
> | L2P                  | SAM[5]               | 83.93                                   | 6.68                                     |
> | **L2P-PGP(ours)**    | **SAM**              | **84.26*****(+0.33)***                        | **5.64*****(-1.04)***   |
> | DualPrompt           | DINO                 | 64.18                                   | 23.87                                    |
> | **DualPrompt(ours)** | **DINO**             | **73.33*****(+9.15)*** | **10.27*****(-13.60)*** |
> | DualPrompt           | SAM                  | 86.11                                   | 6.08                                     |
> | **DualPrompt(ours)** | **SAM**              | **86.92*****(+0.81)***                        | **5.04*****(-1.04)***   |
> | CLIP                 | -                    | 58.95                                   | 6.29                                     |
> | **CLIP-PGP(ours)**   | **-**                | **63.72*****(+4.77)*** | **5.09*****(-1.20)***   |
>
> [1] Zhou, Kaiyang, et al. "Learning to prompt for vision-language models." International Journal of Computer Vision 130.9 (2022): 2337-2348.
>
> [2] Wang, Zifeng, et al. "Learning to prompt for continual learning." Proceedings of the IEEE/CVF Conference on Computer Vision and Pattern Recognition. 2022.
>
> [3] Wang, Zifeng, et al. "Dualprompt: Complementary prompting for rehearsal-free continual learning." European Conference on Computer Vision. Cham: Springer Nature Switzerland, 2022.
>
> [4] DINO pre-trained models (https://arxiv.org/abs/2104.14294).
>
> [5] SAM pre-trained models (https://arxiv.org/abs/2106.01548).

---

> > ### Comment · Reviewer_Q1LH · 2023-11-23
> > **Author response**
> >
> > Thank you for the detailed response. I feel satisfied with the response and will raise the score.

---

> > > ### Author Response · Authors · 2023-11-23
> > > **Thanks for Reviewer Q1LH**
> > >
> > > Thank you for your thoughtful review comments. With your valuable comments and suggestions, we have enhanced the quality of our paper a lot. We sincerely appreciate you and your efforts in reviewing our paper.

---

> ### Author Response · Authors · 2023-11-20
> **Response to Reviewer Q1LH For Question 1**
>
> For **Question 1** about *''Figure 1 is a bit unintuitive as normally''*:
>
> We have **reconfigured** *Figure 1* according to your suggestion and made it more intuitive. We appreciate your valuable feedback which has enhanced the quality of our paper.

---

> ### Author Response · Authors · 2023-11-20
> **Response to Reviewer Q1LH For Question 2**
>
> For **Question 2** about *''modified with gradients for the classifier attached to the backbone''*:
>
> In fact, in prompt-tuning CL, all follow a **special configuration** in the classifier. The details are as follows:
> This framework employs a unified classifier, using the 10-Split-CIFAR-100 dataset as an example. In the first task (including classes from 0 to 9), we train the classifier, and obtain the weights on dimensions of 0-9. In the second task (including classes from 10 to 19), we **freeze** the weights on dimensions of 0-9 in the classifier and continue to train the weights on dimensions of 10-19. We do the same thing in the subsequent tasks. It is obvious that, in this setup, the classifier weights on dimensions of the old classes are not updated **(frozen)**. Therefore, **no gradients are involved**, eliminating the need for gradient update.
>
> Therefore, prompt-tuning CL, **eliminates the need for any projection or alteration of gradients in the classifier**. We have some new results when adopting prototype-based classifiers, but this is beyond the scope of this paper.

---

### Comment · Area_Chair_vhxG · 2023-11-22
**Less than one day**

Dear Reviewers,

If you have already responded to authors rebuttal, Thank you! If not, please take some time, read their responses and acknowledge by replying to the comment. Please also update your score, if applicable.

Thanks everyone for a fruitful, constructive, and respectful review process.

Cheers, Your AC!

---

### Meta-Review · Area_Chair_vhxG · 2023-12-10

**Metareview:**

The paper proposes a novel method that combines prompt tuning with gradient projection to address forgetting in continual learning. It deduces an orthogonal condition for avoiding forgetting in the space of prompt gradients and uses SVD for efficient computation. Experiments show that it reduces forgetting and improves accuracy in various continual learning settings.

Strengths:
- It combines two interesting directions, prompt tuning and gradient projection, in a novel way to tackle forgetting in continual learning.  Beyond continual learning it can pave new ways in transfer or multitask learning with transformer models.
- It provides a sound mathematical derivation of the orthogonal conditions for prompt gradients
- It presents a comprehensive set of experiments to validate the continual learning method across different datasets and continual learning settings.

Weaknesses:
- The setup for some experiments need to be clarified further and   making sure all the baselines have seen fair amount of data and consume a comparable amount of compute.
- For some tasks, limited performance gains over baseline methods is observed which questions the effectiveness of the proposed method considering the added computation and complexity.

What might be missing:
- Comparisons to more recent continual learning methods, especially, projection based methods. To elaborate, both figure 1 and table 1 contains the baselines that are prompt tuning augmented with the projection. To complement this, it would be interesting to start with a projection based and then augment it with prompt tuning.
- The added computation cost for maintaining the orthogonality of the task subspaces can be especially problematic for larger models or and increased number of tasks

**Justification For Why Not Higher Score:**

The paper can still improve with more experiments and baselines (e.g. comparison with gradient-based methods only ad comparison with low rank adaptors). The effectiveness and performance compared to these baselines over larger models or extreme cases of tasks is not verified.

**Justification For Why Not Lower Score:**

The paper presents a novel method, with solid theory and good experimental validation. It can pave a new way in continual and multi-task learning with transformer models.

---

### Decision · Program_Chairs · 2024-01-16

Accept (spotlight)